# Imbalance of stem-like and effector T cell states in children with early type 1 diabetes across conventional and regulatory subsets

Veronika Niederlova [1,2], Ales Neuwirth [1], Vit Neuman [3], Juraj Michalik [1], Bela Charvatova [1], Martin Modrak [4], Zdenek Sumnik [3] & Ondrej Stepanek [1] ✉

Type 1 diabetes (T1D) is an autoimmune disease caused by the loss of self-tolerance toward insulin-producing pancreatic β-cells. Its etiology remains incompletely understood but involves dysregulated T cell responses. Here, we perform single-cell transcriptomic analysis of peripheral blood T cells from children newly diagnosed with T1D, the same children after one year, and healthy donors. We observe that children with diabetes show diminished effector and cytotoxic programs and enhanced stemness-associated gene signature across diverse T cell subsets, especially at diagnosis. In parallel, we detect signs of impaired regulatory capacity in regulatory T cells and regulatory TR3-56 cells. These findings are supported by flow cytometry analysis of the same cohort and reanalysis of publicly available datasets. Overall, our results suggest that T1D is associated with impaired T cell effector differentiation and regulatory T cell dysfunction, both of which may contribute to immune imbalance and loss of self-tolerance.

Type 1 diabetes mellitus (T1D) is an autoimmune disease characterized by the destruction of pancreatic β-cells, leading to insulin deficiency, hyperglycemia, and impaired metabolic homeostasis. As there is currently no cure for T1D, people with T1D rely on insulin replacement therapy for the rest of their lives. Unlike most autoimmune diseases, T1D typically manifests in childhood. Both genetic and environmental factors contribute to the development of T1D. The genetic factors include specific HLA haplotypes[1], polymorphisms in the insulin gene (*INS*)[2] and several immune-related genes (e.g., *CTLA4*, *PTPN22*, *IFIH1*, and *CD226*)[3]. The importance of the environmental factors is highlighted by an increase in T1D incidence in high-income countries[4–6]. It has been proposed that specific viral infections, such as enteroviruses, can trigger T1D[7,8]. On the contrary, the hygiene hypothesis proposes that the reduction of the infection burden in children may partly explain the rising incidence of T1D and other autoimmune diseases[4,9–11]. Despite some controversies in this area of research, it is well-established that a genetically and/or environmentally altered state of the immune system is a significant risk factor for T1D and may

contribute directly to its onset. However, the etiology of T1D is still largely unknown.

T cells are extensively studied in the context of T1D because of the genetic evidence of HLA involvement, their clear role in the development of T1D in animal models[12,13], and their presumed role in β-cell destruction via cell-mediated cytotoxicity[14,15]. Moreover, it has been proposed that T1D susceptibility may be induced by insufficient self-tolerance mediated by FOXP3+ regulatory T cells (Treg) and non-classical regulatory T cell populations, such as CD3+ CD56+ TR3-56 cells[16]. Despite numerous studies in the field[17–23], the T1D-associated alterations in the T cell compartment are still not fully understood.

Traditionally, the immune cells in T1D are studied using targeted flow cytometry panels or bulk transcriptomics on whole blood, peripheral blood mononuclear cells (PBMC), and/or sorted leukocyte subsets[24–28]. Whereas flow cytometry is limited to the pre-selected protein markers, bulk transcriptomic analysis lacks flow cytometry's single-cell resolution. These limitations can be overcome by multi-parameter approaches on the single-cell level, such as cytometry by

[1]Laboratory of Adaptive Immunity, Institute of Molecular Genetics of the Czech Academy of Sciences, Prague, Czechia. [2]Department of Cell Biology, Faculty of Science, Charles University, Prague, Czechia. [3]Department of Pediatrics, 2nd Faculty of Medicine, Charles University & Motol University Hospital, Prague, Czechia. [4]Department of Bioinformatics, 2nd Faculty of Medicine, Charles University, Prague, Czechia. ✉e-mail: ondrej.stepanek@img.cas.cz

time-of-flight or single-cell RNA sequencing (scRNAseq). Indeed, pioneering studies established these techniques as the methods of choice to capture the differences between the immune cells in children with and without T1D[21,26,29,30]. However, these studies also revealed potential caveats, such as a trade-off between the sufficient size of the cohort to address reproducibility and the number of sequenced cells per donor to achieve sufficient resolution[26,29,30]. In parallel efforts, big consortia such as The Network for Pancreatic Organ Donors with Diabetes (nPOD)[31] and Human Pancreas Analysis Program (HPAP)[22,32] are generating large atlases of multimodal data from tissues of deceased donors, including scRNAseq, flow cytometry, immune repertoire profiling, and imaging data. Although these efforts provide invaluable data sets, including samples from solid organs that are hardly accessible, their disadvantage is their inability to recruit donors specifically based on features such as their age or time after diagnosis.

In this study, we compare the T cell compartments in children with and without T1D using single-cell transcriptomics, and we monitor longitudinal changes during the first year after diagnosis. T cells from newly diagnosed children display a bias toward naïve-like and stem-associated states, which is partially normalized after one year. In addition, Tregs show impaired functional differentiation with a gene signature of dysfunction. Together, these findings highlight a disrupted balance between effector and regulatory programs in T cells, providing new insights into mechanisms that may underlie loss of self-tolerance in T1D.

## Results

### Generation of a T cell atlas of children with T1D and healthy donors

We performed a prospective study on a cohort of 43 pediatric donors with T1D or without T1D (henceforth referred to as healthy donors) using scRNAseq to identify potential gene expression patterns and heterogeneity in blood T cells associated with T1D. Specifically, we searched for differences in the T cell compartment between the healthy donors and children with T1D, and for changes occurring during the first year after the diagnosis of T1D. We based our analysis on the scRNAseq, including T cell receptor sequencing (VDJ profiling) and flow cytometry analysis of T cells from the peripheral blood of our Czech cohort, complemented by the analysis of openly accessible data from other cohorts.

Blood samples from 30 children with T1D were taken 4–24 (median 9) days after their diagnosis (time T0) and again at 359–423 (median 377) days after diagnosis (time T1). For comparison, we collected blood from 13 healthy donors (Fig. 1A). The T1D and healthy donor cohorts were comparable in age distribution, as confirmed by descriptive statistics (Supplementary Table 1, Supplementary Fig. 1A) and standardized mean difference (SMD) analysis (Supplementary Fig. 1B). However, the sex distribution differed, with a higher proportion of males in the T1D cohort (67%) compared to the healthy donors (33%) (Supplementary Table 2). We characterized the donors by their age, sex, partial clinical remission status at T1, T1D-associated autoantibody levels, HLA haplotypes (Fig. 1B), as well as blood concentrations of glycated hemoglobin, random and fasting C-peptide, and leukocyte levels (Supplementary Fig. 1C–E). At T0, children with T1D showed a higher proportion of lymphocytes (mean 48.1%, SD 12.9%) and a lower proportion of neutrophils (mean 38.9%, SD 13.1%) compared to healthy donors (lymphocytes: mean 40.4%, SD 11.0%; neutrophils: mean 46.5%, SD 12.6%) and to their own levels at T1 (lymphocytes: mean 37.9%, SD 11.2%; neutrophils: mean 49.2%, SD 11.8%) (Supplementary Fig. 1E).

First, we performed an initial scRNAseq analysis of CD4[+] and CD8[+] lymphocytes from 16 samples, which revealed that the majority of T cells in our child donors had a naïve CD45RA[+] CD45RO[-] phenotype (Supplementary Fig. 2A–D), determined by CD45 splicing-sensitive analysis tool IDEIS[33]. To enable a more detailed characterization of

antigen-experienced (non-naïve) T cells in the final scRNA-seq analysis, we enriched for these cells by sorting at a fixed 1:5 ratio of naïve to antigen-experienced cells for both CD4[+] and CD8α[+] lymphocytes (Supplementary Fig. 2E–H). Details on sample counts, donor numbers, and enrichment strategies for both initial and final experiments are summarized in Supplementary Table 3.

In total, we processed 176 samples, with an average of 992 cells per sample. On average, each cell contained 2689 non-redundant transcripts and 1464 detected genes (Supplementary Table 4). We processed the data using our standard pipeline[34] including the removal of dead cells and doublets (Supplementary Fig. 3A, B). Because the library preparation was performed in six batches, we used integration by STACAS to remove the batch effect[35,36]. Integration successfully removed the unwanted variability between experiments, but importantly, it preserved biologically relevant differences, such as the difference between initial and final experiments, which contained non-enriched versus enriched cells (Supplementary Fig. 4A–H). Comparison of data from initial and final experiments showed that, as expected, our approach reduced the frequency of naïve cells from ~ 70–75% to 20–25% (Supplementary Fig. 4I–J). All subsequent analyses were restricted to the final experiments, i.e., enriched samples, to ensure consistency.

In the next step, we generated CD8[+] and CD4[+] lymphocyte atlases using the processed scRNAseq data. The CD8[+] lymphocyte compartment consisted of three clusters: CD8α[low] NK cells, unconventional CD8[+] T cells represented by semi-invariant MAIT cells and γδT cells, as well as conventional CD8[+] αβT cells (Supplementary Fig. 5A–D). Based on the typical markers, we identified that conventional CD8[+] T cells contain naïve, memory (Tmem), unconventional KLRC2[+] memory, also previously described as KILR-like cells[37,38], which we abbreviate as Tmk (T cell memory KLRC2[+]), terminally differentiated CD45RA[+] (Temra), and proliferating subsets (Fig. 1C, D and Supplementary Fig. 5E). Temra and to a lesser extent, proliferating and Tmem cells, but not Tmk cells, were clonally expanded (Fig. 1E). These subsets could be further subdivided into 41 smaller subsets (Supplementary Fig. 5F). The identity of cell subsets was validated using a previously published RNAseq dataset of sorted populations of CD8[+] T cells[39] (Supplementary Fig. 5G, H).

The CD4[+] lymphocyte compartment consisted of a subset of unconventional T cells including iNKT cells expressing their signature TCRα (*TRAV10, TRAJ18*; CDR3: CVVSDRGSTLGRLYF) (Supplementary Fig. 6A, B) and conventional CD4[+] T cells, which could be further divided to naïve, central memory (Tcm), Treg, Th1/Th17, Th2, Temra, CD4[+] T cells with a high expression of interferon signaling associated genes (ISAGhi)[40], proliferating cells, and CD4[+] T cells enriched for genes involved in signal transduction pathways (Tsig) (Fig. 1F, G and Supplementary Fig. 6C–F). Temra cells were the only CD4[+] T cell subset showing a substantial level of clonal expansion (Fig. 1H). These subsets could be further split into 38 smaller subsets (Supplementary Fig. 6G). The identity of cell subsets was validated using a previously published RNAseq dataset[39] (Supplementary Fig. 6H, I).

We compared the proportions of particular subsets using both frequentist (Supplementary Fig. 7A–D) and Bayesian (Supplementary Fig. 7E, F) statistics. These analyses did not reveal substantial differences between children with T1D and healthy donors, with the exception of CD8[+] Temra cells, which were enriched in the healthy donors in comparison to the children with T1D (Supplementary Fig. 7A, E). The Bayesian statistics revealed apparent differences in ISAGhi and Tsig subsets (Supplementary Fig. 7F), which, however, originated from only a limited number of outlier participants (Supplementary Fig. 7C). Within the T1D group, the frequency of MAIT cells among CD8[+] T cells and the frequency of Th2 cells among CD4[+] T cells correlated with fasting C-peptide level, which we used as a proxy for residual β-cell activity (Supplementary Fig. 7G, H). However, because of the prior enrichment for antigen-experienced T cells, the relative abundance of the particular subsets in the scRNAseq data does not correspond to the real frequency in the original sample.

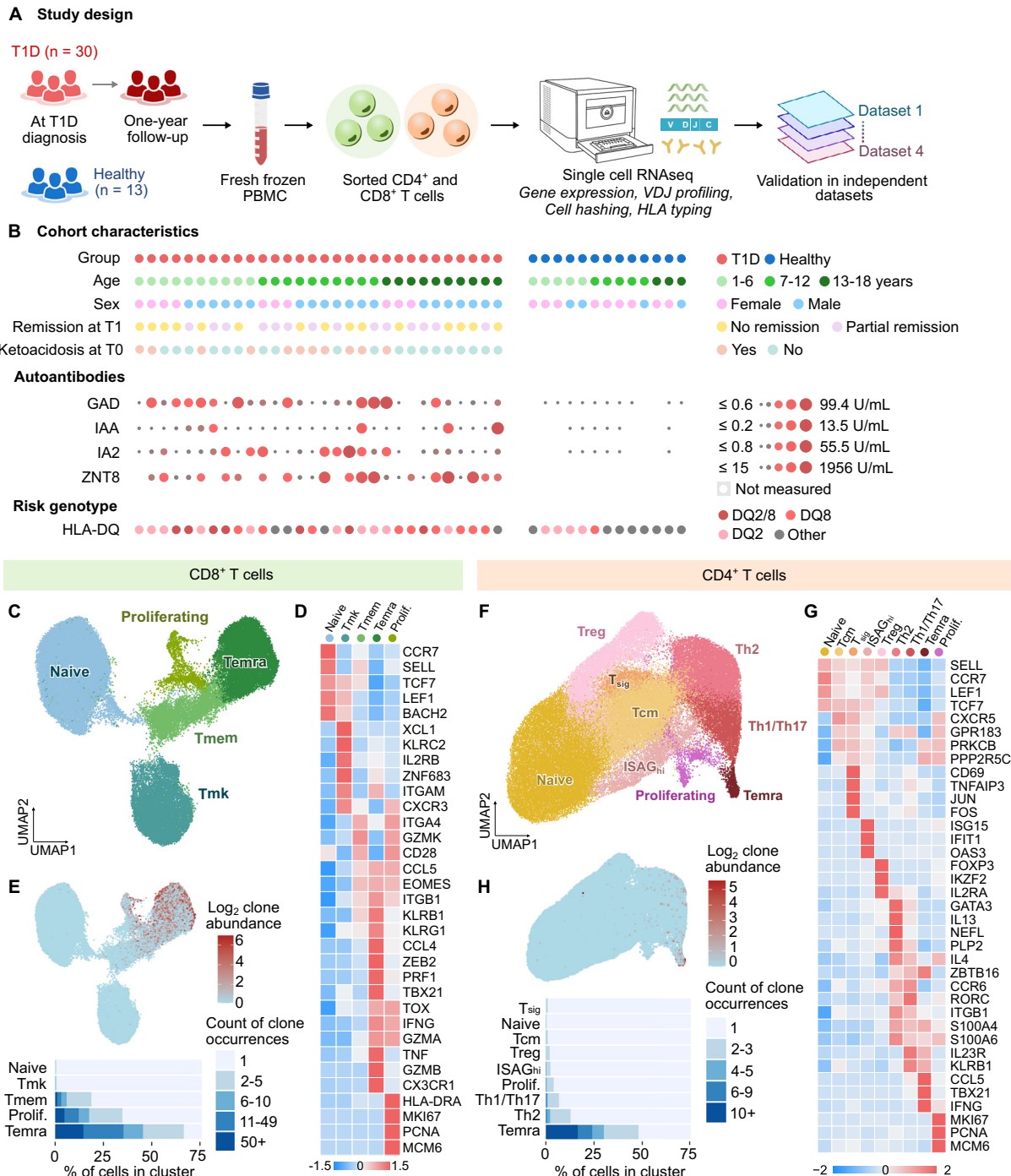

**Fig. 1 | Blood T cell atlas of children with T1D and healthy donors.**
**A** Experimental design. PBMCs from children with newly diagnosed T1D (T0, $n = 30$), one-year follow-up (T1, $n = 29$), and age-matched healthy donors (HD, $n = 13$) were collected and cryopreserved. Sorted CD4$^+$ and CD8$^+$ T cells underwent scRNA-seq (gene expression and paired TCR repertoire profiling). Findings were validated in independent published cohorts. **B** Cohort characteristics. Each dot indicates one donor. Upper panel: group, age, sex, remission at T1 (defined as insulin dose–adjusted HbA1c < 9) and diabetic ketoacidosis at T0 (defined as pH < 7.3 and/or bicarbonate < 15 mEq/L). Middle panel: autoantibody status (GAD, IAA, IA2, ZNT8). Lower panel: T1D risk HLA alleles inferred from scRNA-seq. **C–E** CD8$^+$ T cells ($n = 63,068$ cells from 43 donors) after quality control and exclusion of contaminants, unconventional subsets, and CD8low NK cells. **C** UMAP of conventional CD8$^+$ T cell populations. Louvain clusters were merged by functional similarity and annotated by marker genes. **D** Heatmap of selected cluster-defining

genes (row-scaled z-scores of average expression). **E** TCR repertoire analysis: UMAP colored by clonal expansion (log$_2$ counts of recurrent paired CDR3α/CDR3β). Bottom: quantification of expansion across clusters. **F–H** CD4$^+$ T cells ($n = 79,876$ cells from 43 donors) after exclusion of low-quality, contaminating, and unconventional subsets. **F** UMAP of conventional CD4$^+$ T cell populations. Louvain clusters merged and annotated as in (**C**). **G** Same as (**D**) but for CD4$^+$ clusters. **H** TCR repertoire analysis: UMAP colored by clonal expansion, calculated as in (**E**), with quantification across clusters. T1D – Type 1 Diabetes Mellitus, T0 – timepoint at diagnosis, T1 – timepoint one year after diagnosis, Tmk – unconventional KLRC2$^+$ memory T cells, Tmem – memory T cells, Tcm – central memory T cells, Treg – regulatory T cells, Tsig – T cells with signaling transduction signature, ISAGhi – T cells with interferon signaling signature, GAD – Glutamic Acid Decarboxylase, IAA – Insulin Autoantibodies, IA2 – Tyrosine Phosphatase-like Protein IA-2, ZNT8 – Zinc Transporter 8.

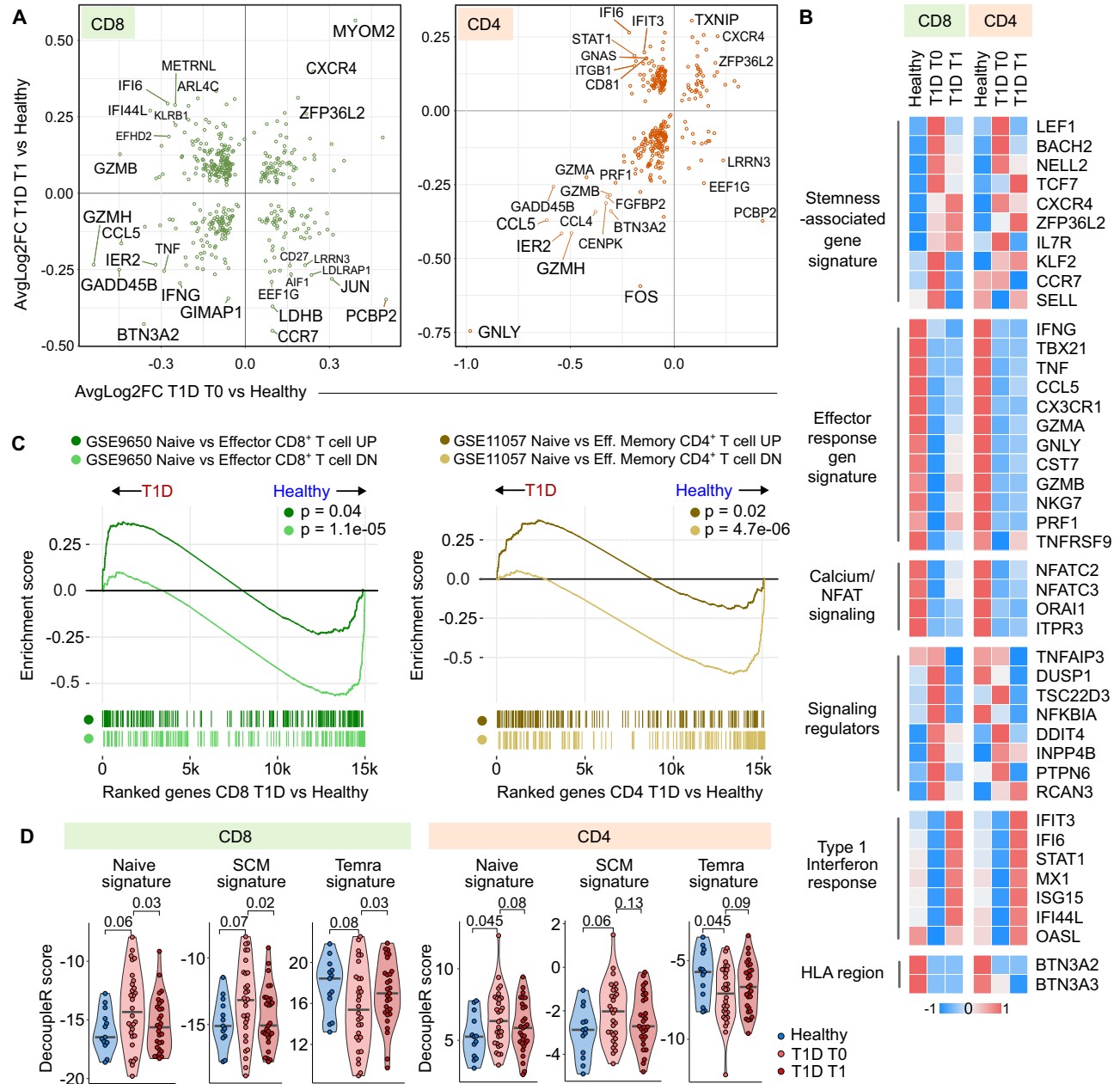

**Fig. 2 | Differential gene expression in T cells in children with T1D.**
**A** Differentially expressed genes (DEGs) in CD8+ (left) and CD4+ (right) T cells comparing children with T1D at diagnosis (T0, *n* = 30) or one year after diagnosis (T1, *n* = 29) with healthy donors (HD, *n* = 13). Each point represents one gene, plotted by average log₂ fold change in T1D T0 vs. HD (*x*-axis) and T1D T1 vs. HD (*y*-axis). DEGs were identified using the two-sided Wilcoxon rank-sum test (Seurat FindMarkers), with *p*-values adjusted for multiple comparisons (Bonferroni). Only genes with adjusted *p* < 0.05 are shown; selected top up- and downregulated genes are labeled. **B** Heatmap of relative expression of selected DEGs across HD, T1D T0, and T1D T1. Donor-level averages were calculated using Seurat AverageExpression, z-score normalized within each gene, and then averaged per group. **C** Gene set enrichment analysis (GSEA) showing enrichment of naïve versus effector T cell signatures from published datasets (GSE9650, GSE11057). Each curve corresponds to genes upregulated (dark, "UP" set) or downregulated (light, "DN" set) in naïve versus effector cells. Positive normalized enrichment scores indicate enrichment of the naïve-up gene set and depletion of effector-up genes in T1D samples. *P*-values were estimated using an adaptive multi-level split Monte Carlo scheme (fgsea R package). **D** Violin plots of decoupleR scores quantifying similarity to reference contrasts (Temra vs. naïve; naïve vs. Tem) in CD8+ (left) and CD4+ (right) T cells. Each dot = one donor. Comparisons between HD and T1D T0 were assessed using two-tailed Mann–Whitney tests; paired comparisons between T1D T0 and T1 used two-tailed paired Mann–Whitney tests. Bars indicate medians. *n* = 13 HD, 30 T1D T0, 29 T1D T1. T1D – Type 1 Diabetes Mellitus, T0 – timepoint at diagnosis, T1 – timepoint one year after diagnosis, GSEA – Gene set enrichment analysis.

## The onset of diabetes is associated with low effector T cell signatures

Next, we compared gene expression between healthy donors and children with T1D at T0 and T1, separately in CD4+ and CD8+ T cells (Fig. 2A, B). We visualized the consistency of gene expression changes by plotting log-fold changes in T1D T0 vs. healthy donors against T1D

T1 vs. healthy donors for each gene, showing the key differences at both timepoints (Fig. 2A).

At both timepoints, we found an enrichment of effector and cytotoxic genes (*GZMA, GZMB, GZMH, PRF1, GNLY, IFNG, TNF, CCL5, CCL4*) in healthy donors compared to T1D. In contrast, genes associated with naïve, quiescent, and/or stem-like T cells (*TCF7, LEF1,*

*BACH2, IL7R, CXCR4, ZFP36L2*) were upregulated in children with T1D. Hereafter, we refer to the latter set of genes as a stemness-associated gene signature, reflecting transcriptional features shared by naïve, quiescent, and stem memory T cells, rather than being strictly limited to canonical naïve cells. This bias toward a stemness-associated signature was most pronounced at T0 (Fig. 2B).

T cells from T1D T0 samples also showed higher expression of several regulators of intracellular activation signaling pathways (*DUSP1, TSC22D3, TNFAIP3*), which might be associated with the quiescent state (Fig. 2B). In contrast, samples from healthy donors showed a higher expression of genes associated with calcium signaling and NFAT activation (*ORAI1, NFATC2, NFATC3*), potentially supporting a more activated, effector-like state.

To further support these observations, we performed gene set enrichment analysis (GSEA) using signatures of genes upregulated or downregulated in naïve versus effector or effector memory T cells from previously published datasets (GSE9650[41], GSE11057[42]). This analysis confirmed the enrichment of the naïve (stemness-associated) signature in T cells from children with T1D and an effector signature in T cells from healthy donors (Fig. 2C).

To validate these findings at the sample level, we performed differential expression analysis using DESeq2 on pseudobulk expression profiles aggregated per donor. This analysis confirmed a strong correlation of log-fold changes with the original per-cell analysis (Supplementary Fig. 8A, B). Key genes representing effector-, stemness-, and NFAT-associated signatures were identified by both types of analysis (Supplementary Fig. 8C–F).

In addition, for each donor, we quantified the similarity of their gene expression to reference contrasts of naïve versus memory cells from a previously published dataset (GSE179613[39]) using decoupleR[43], which, unlike GSEA, explicitly accounts for the sign and weight of network interactions. This comparison revealed enrichment of naïve and stem memory signatures in T1D samples at T0, and enrichment of Temra signatures in healthy donors (Fig. 2D). These patterns were observed across different age and sex groups, confirming that these differences are not driven by age or sex (Supplementary Fig. 9A, B).

In addition to the bulk CD4$^+$ and CD8$^+$ T cell compartments, we also analyzed differential expression within individual T cell subsets (Supplementary Figs. 10A, B, 11A, B). This revealed that while most genes followed similar patterns across subsets, certain differentially expressed genes (DEG) (e.g., *MYOM2, IFI27, CD38, GZMK, FOS, FOSB*) showed distinct, subset-specific regulation. To quantify these overlaps, we assessed shared top DEGs across subsets, showing high similarity among naïve and memory populations (Supplementary Figs. 10B, 11B).

To check whether the downregulation of effector genes and upregulation of naïve genes in children with T1D is associated with the disease severity, we carried out the comparison of differential expression between children with T1D at T0 who suffered from ketoacidosis and those who did not, and between T1D who fulfilled or not the criterion of partial clinical remission (insulin dose adjusted HbA1c < 9[44,45]) (Supplementary Fig. 12A–D). We did not observe a pronounced bias towards stem-like or effector-like gene expression in either of these subgroups (Supplementary Fig. 13A, B).

For the interpretation of the previous results, it should be noted that the ratio of naïve and antigen-experienced T cells was normalized for the scRNAseq analysis. For this reason, we analyzed the samples by flow cytometry to reveal that both antigen-experienced CD8$^+$ and CD4$^+$ T cells were less abundant in children with T1D than in healthy donors at T0, but not at T1 (Fig. 3A). Accordingly, the expression of a key cytotoxic protein Granzyme B (GZMB) is lower in children with T1D in CD8$^+$ and CD4$^+$ T cells than in the healthy donors at both time points (Fig. 3A). The frequencies of populations analyzed by scRNAseq before enrichment correlated with the frequencies obtained by flow cytometry, indicating the consistency of the two methods (Supplementary Fig. 14A).

Overall, the data suggested that the T cell compartment has a more stem-like and less effector-like state in children with T1D than in healthy donors. This stem-like state is partially normalized within one year after the diagnosis and the associated introduction of the insulin therapy.

To extend our findings to unrelated T1D cohorts, we addressed the question of the stem-like vs. effector signature in people with T1D in previously published data. Using the HPAP database[32], we re-analyzed scRNAseq data obtained from T cells from the spleens of 53 children and adult donors with or without T1D. In this cohort, the effector signature genes were enriched in the control samples, and the stemness-associated (naïve-like) signature genes were enriched in donors with T1D (Fig. 3B). In addition, we re-analyzed flow cytometry data from the HPAP and observed a reduction of effector CD8$^+$ and CD4$^+$ T cells in the blood of donors with T1D (Fig. 3C and Supplementary Fig. 14B), consistent with the findings in our pediatric cohort.

Next, we collected transcriptomic data from previously published studies[26,29,30,46,47] (Supplementary Table 5) and re-analyzed it using the same pipeline. Genes consistently upregulated across T1D datasets included those connected to the naïve, stem-like, and quiescent T cell states such as *LEF1*[48], *FOXP1*[49], *BACH2*[50], *NELL2*[51], *IKZF1*[52] and a negative regulator PI3 kinase signaling *INPP4B*[53] (Fig. 3D). In contrast, genes associated with cytotoxicity and effector functions, such as *NKG7, GNLY, CCL4, CCL5, CX3CR1, TNFRSF4* (OX40)*, TNF*, and *TNFRSF9* (4-1BB) were upregulated in healthy donors. Altogether, our results showing a T cell bias towards a naïve-like, stem-associated, and quiescent state in T1D are supported by the majority of the published datasets.

## BTN3A2 expression depends on HLA haplotypes

It was shown that *BTN3A2* and to a lower extent, *BTN3A3*, encoding butyrophilin family members, are upregulated in T cells of children with T1D compared to healthy donors[26]. However, we observed the opposite in our dataset, i.e., the higher expression of *BTN3A2* and *BTN3A3* in healthy donors than in children with T1D (Fig. 2A, B). Since *BTN3A2* is a part of the extended MHC region[54], we hypothesized that its expression may depend on specific HLA haplotypes rather than T1D status.

In our cohort, most children with T1D carried known susceptibility alleles, specifically HLA-DR3-DQ2 (haplotype HLA-DRB1 03:01–DQA1 05:01–DQB1 02:01) and/or HLA-DR4-DQ8 (HLA-DRB1 04:01–DQA1 03:01–DQB1 03:02)[1], while healthy donors reflected the general HLA distribution in the Czech population[55] (Supplementary Fig. 15A). We used our data and publicly available data to estimate the role of the T1D status, particular haplotype, and study on *BTN3A2* expression. We observed that the study and HLA haplotypes, but not the T1D status, are the factors significantly influencing *BTN3A2* expression (Supplementary Figs. 15B, 16). This indicates that the selection of healthy donors (e.g., unmatched as in our study or partially matched as in ref. 26) might have a great impact on the results concerning *BTN3A2* expression.

## Naïve-like regulatory T cells in children with T1D

It has been proposed that the dysfunction of Treg cells might contribute to the loss of self-tolerance and the development of autoimmune diabetes in humans and animal models[56,57]. We identified four Treg subclusters labeled Treg1 to Treg4 (Fig. 4A). The gene expression analysis revealed that Treg1 represents Tregs with the most naïve/stem-like phenotype, with high expression of stemness-associated genes such as *CCR7, TCF7, SELL* and relatively low expression of Treg signature genes such as *FOXP3, CTLA4*, and *IL2RA* (Fig. 4B–D). However, the expression of these signature molecules in Treg1 cells was higher than in naïve T cells suggesting that they are bona fide Tregs (Fig. 4D). In contrast, the Treg4 subset represents Tregs with the most activated phenotype, which predispose them for potent regulatory functions[58,59].

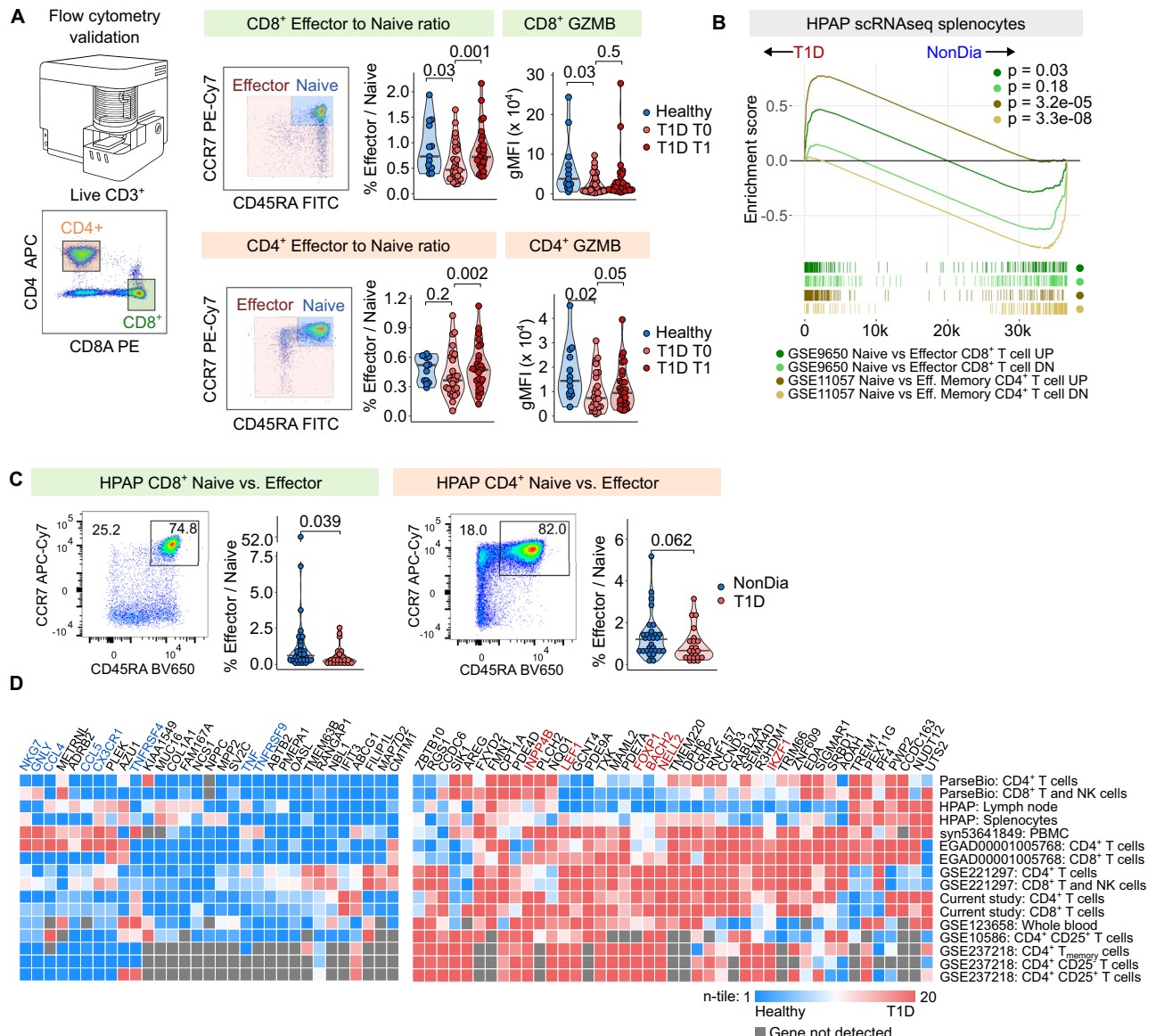

**Fig. 3 | Validation of scRNAseq results by flow cytometry and reanalysis of published datasets. A** Validation by flow cytometry. Effector and naïve T cells were gated from live CD3⁺ CD8⁺ or CD4⁺ T cells. For CD8⁺ (left) and CD4⁺ (right) T cells, plots show (left to right): representative gating of naïve (CCR7⁺CD45RA⁺) vs. effector populations, effector-to-naïve ratio, and geometric mean intensity of GZMB. Full gating in Supplementary Fig. 2. Comparisons between HD and T1D T0 were assessed using two-tailed Mann–Whitney tests; paired comparisons between T1D T0 and T1 used two-tailed paired Mann–Whitney tests. Bar at median. $n = 13$ healthy donors, $n = 30$ T1D T0 donors, $n = 29$ T1D T1 donors. **B** Gene set enrichment analysis (GSEA) of DEGs identified in published datasets (GSE9650, GSE11057) using splenocytes from T1D and non-diabetic donors in the HPAP dataset. *P*-values estimated by the adaptive multi-level split Monte Carlo scheme (fgsea R package). **C** Flow cytometry of CD8⁺ (left) and CD4⁺ (right) T cells from PBMCs of T1D and

non-diabetic (NonDia) HPAP donors. Plots show (left to right): representative naïve vs. effector gating and violin plots for quantification. Full gating in Supplementary Fig. 14. *P*-values by two-tailed Mann–Whitney test; bars at medians. $n = 32$ NonDia, $n = 23$ T1D. **D** Heatmap of DEGs consistently dysregulated in T1D vs. healthy donors across published studies. Transcriptomic count matrices from multiple datasets (Supplementary Table 5) were processed using a unified pipeline. Fold changes (T1D vs. healthy) were calculated for all genes, then divided into 20 quantiles (quantile 1 = most upregulated in healthy; quantile 20 = most upregulated in T1D). Genes most consistently enriched in quantiles 1 (left) and 20 (right) across studies are shown. HPAP – Human Pancreas Analysis Program, T1D – Type 1 Diabetes Mellitus, T0 – timepoint at diagnosis, T1 – timepoint one year after diagnosis, GSEA – Gene set enrichment analysis.

The naïve-like Treg subset was more abundant among Tregs in the children with T1D than in healthy donors (Fig. 4E, F and Supplementary Fig. 17A) at T0 and T1. Treg1 cells expressed high levels of *CD226* and low levels of *TIGIT* (Fig. 4C), which both bind to common ligands CD112 and CD155, but have the opposite functions. CD226 is associated with Treg dysfunction, whereas TIGIT in Tregs is linked to the self-tolerance in the context of T1D[60–63].

To further understand the relationship among Treg subsets, we performed pseudotime analysis[64], which revealed a trajectory

consistent with progressive Treg activation and differentiation (Fig. 4G). Pseudotime strongly correlated with a Treg activation score (Fig. 4H), and median pseudotime values indicated that Tregs were more activated in healthy donors compared to children with T1D (Fig. 4I). To formally test for differences in trajectory progression, we applied the condiments package[65], which confirmed significant differences in pseudotime dynamics between groups (Fig. 4J–L).

In the next step, we addressed whether Tregs from T1D resemble functionally impaired Tregs from conditions in which their

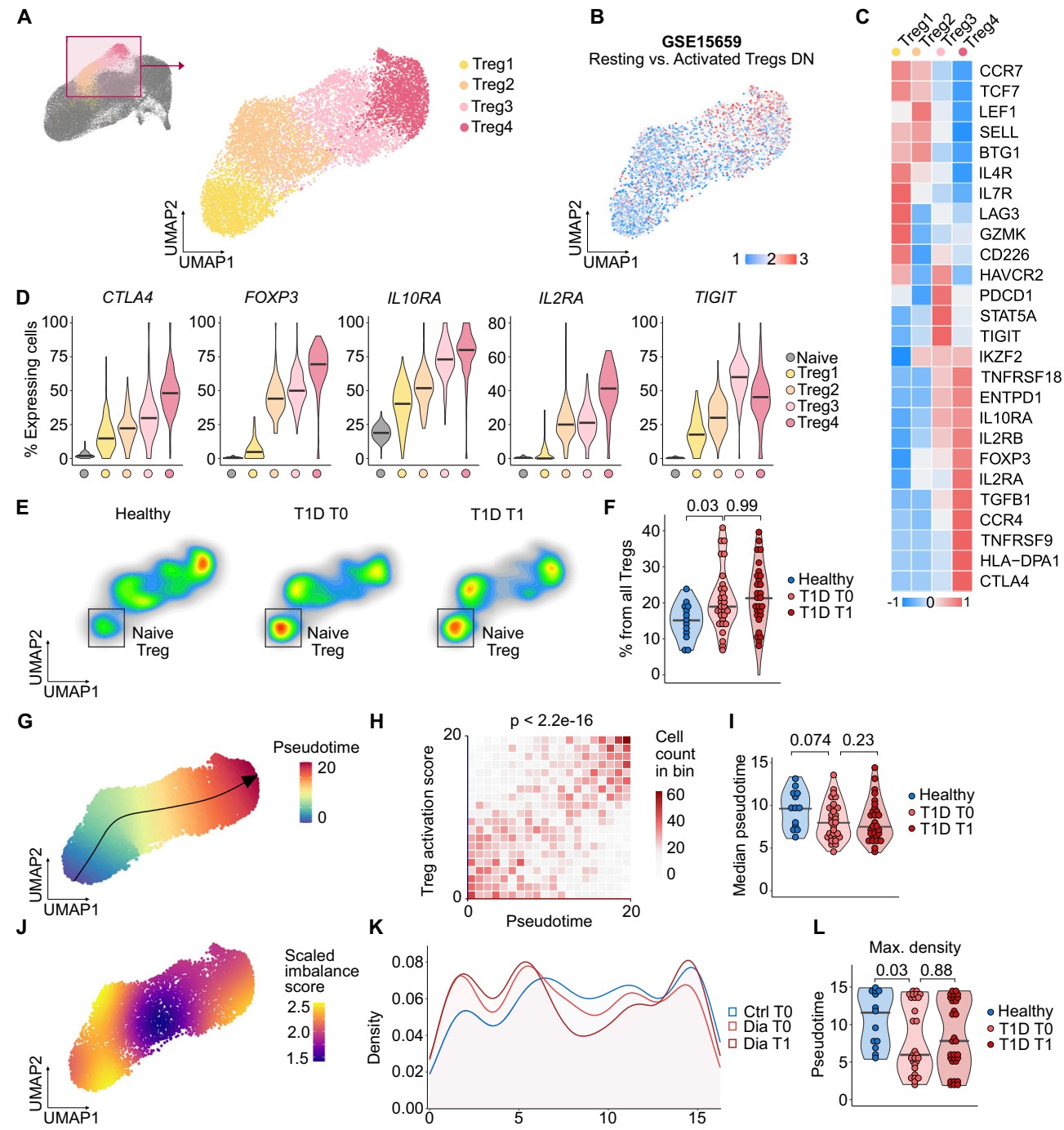

**Fig. 4 | Regulatory T cells in children with T1D. A** Reclustering of Treg clusters from the UMAP projection shown in Fig. 1F. Selected clusters were extracted and subjected to new normalization, scaling, integration, and dimensionality reduction. The resulting UMAP projection is shown on the right. $n = 9890$ cells from 43 donors. Cells are colored by cluster. **B** Same UMAP projection as in (**A**), colored by expression of a gene module upregulated in activated Tregs compared to resting Tregs from GSE15659[58]. **C** Heatmap of selected marker genes characterizing the clusters shown in (**A**). Color represents row-scaled z-score of average gene expression per cluster. **D** Violin plots of the percentage of cells per cluster with non-zero expression of selected genes (87 samples). Bars show medians. **E** Density plots of Treg subtype abundance across healthy donors and children with T1D at T0 and T1. **F** Quantification of the percentage of naïve Tregs per sample among total Tregs, gated as shown in (**E**). P-values were calculated using a two-tailed Mann−Whitney test (healthy donors vs. children with T1D at T0) or a two-tailed paired Mann−Whitney test (children with T1D at T0 vs. T1). Bars indicate medians. $n = 13$

healthy donors, $n = 30$ T1D T0, $n = 29$ T1D T1. **G** Same UMAP as (**A**), colored by pseudotime values calculated using the slingshot package. The arrow indicates the identified lineage trajectory. **H** Tile plot of correlation between pseudotime and Treg activation score from (**B**), binned into 20 intervals. P-value by two-sided Pearson correlation. **I** Violin plots showing median pseudotime values in healthy donors and children with T1D at T0 and T1. P-values as in (**F**). Bars indicate medians. $n = 13$ healthy donors, $n = 30$ T1D T0, $n = 29$ T1D T1. **J** Same UMAP as (**A**), colored by scaled imbalance score estimated using the condiments package. Higher scores indicate greater differences in cell density between healthy donors and children with T1D at T0 or T1. **K** Density of cells from healthy, T1D T0, and T1 along pseudotime. **L** Violin plots showing the pseudotime value at which cells show maximum density in each condition. P-values as in (**F**). Bars indicate medians. $n = 13$ healthy donors, $n = 30$ T1D T0, $n = 29$ T1D T1. T1D – Type 1 Diabetes Mellitus, T0 – timepoint at diagnosis, T1 – timepoint one year after diagnosis.

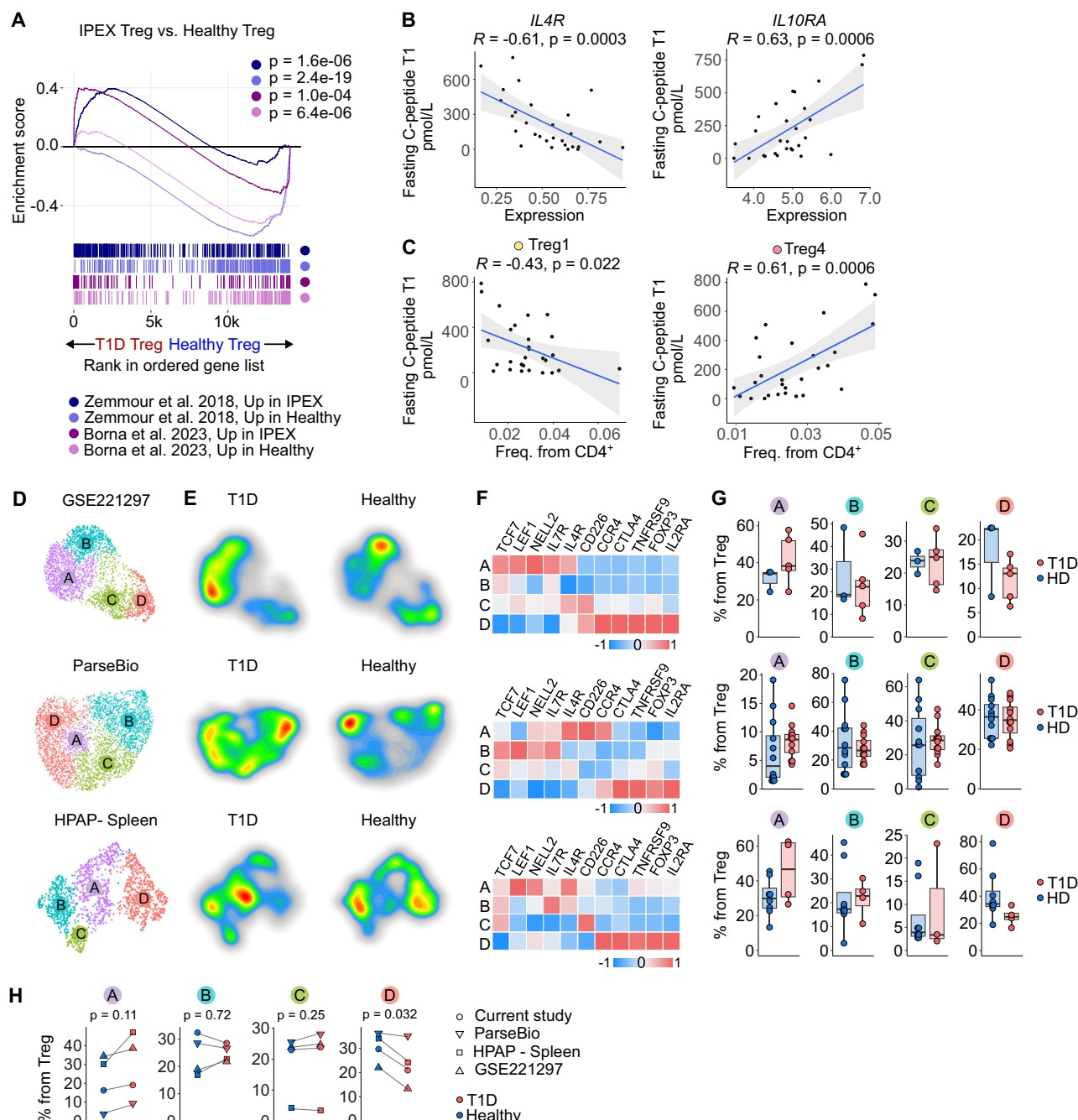

**Fig. 5 | Altered Treg signatures in children with T1D and validation in independent datasets. A** Gene set enrichment analysis (GSEA) of genes up- or down-regulated in IPEX patients vs. healthy donors (refs. 65,66) tested against ranked DEGs in Tregs from children with T1D vs. healthy donors. *P*-values estimated using adaptive multi-level split Monte Carlo (fgsea R package). **B** Correlation between fasting C-peptide at T1 and IL4R (left) or IL10RA (right) expression within Tregs at T0. *P*-values from two-sided Pearson correlation; *n* = 28 T1D donors. Line = regression fit; shaded area = 95% CI. **C** Correlation between fasting C-peptide at T1 and frequency of Treg1 (left) or Treg4 (right) at T0. *P*-values, line, and shaded area as in (**B**); *n* = 28 T1D donors. **D–G** Reanalysis of Treg populations in published datasets: GSE221297 (PBMC, 5 T1D, 3 healthy), ParseBio (PBMC, 12 T1D, 12 healthy),

and HPAP (splenocytes, 4 T1D, 8 non-diabetic). **D** UMAP projection of Treg cells, colored by Louvain clusters. **E** Density plots showing distribution differences of Tregs from healthy vs. T1D donors. **F** Heatmaps of selected marker genes per cluster (row-scaled z-scores of average expression). **G** Quantification of cluster frequencies per sample. Boxplots: line = median, hinges = first and third quartiles, whiskers = 1.5 × IQR. **H** Average frequency of Treg subclusters in healthy (blue) and T1D (red) donors across datasets. Clusters Treg1–4 in our study matched to TregA–D here. *P*-values by paired *t* test. HD – Healthy Donor, IPEX - Immuno-dysregulation polyendocrinopathy enteropathy X-linked syndrome, T1D – Type 1 Diabetes Mellitus, T1 – timepoint one year after diagnosis.

dysfunction is well established and characterized. First, we compared T1D Tregs from our cohort to Tregs from patients with a disease called *Immune dysregulation, polyendocrinopathy, enteropathy X-linked syndrome* (IPEX), which is caused by loss-of-function mutations in *FOXP3*. Tregs from children with T1D upregulated genes specific for defective

Treg-like cells observed in IPEX patients[66,67] (Fig. 5A). We obtained the same results when we compared Tregs from children with T1D to Treg-like cells in *Foxp3*[-/-] mice[68,69] and bona fide human Tregs after their key transcriptional regulators *FOXP3, HIVEP2, IKZF2* (Helios), or *SATB1* were knocked-out[70] (Supplementary Fig. 17B, C). Moreover, we observed an

inverse correlation between *IL4R* expression in Tregs and residual insulin production measured as fasting C-peptide levels (Fig. 5B), which points to a previously proposed inhibitory role of IL-4 signaling in Treg differentiation[71,72]. In contrast, the expression of *IL10RA* in Tregs correlated with fasting C-peptide levels, which highlights a possible role of IL-10 signaling for Treg-mediated tolerance in T1D (Fig. 5B). Fasting C-peptide levels also correlated with the frequency of Treg1 cells (negative correlation) and the frequency of Treg4 cells (positive correlation) (Fig. 5C), suggesting that the Treg maturation status might be involved in the disease severity. Overall, this evidence suggests that Tregs in children with T1D might be less mature and/or less functional than their counterparts in healthy donors.

To address our findings in independent cohorts, we reanalyzed data from three publicly available scRNAseq datasets studying peripheral immune cells of donors with T1D and healthy donors: a recent publication[30], HPAP dataset[32], and a reference dataset published by Parse Biosciences (henceforth ParseBio). In each dataset, we identified Treg cells based on canonical markers, then extracted and reclustered these cells to define four clusters (A–D), which roughly corresponded to our Treg1–4 clusters based on their gene expression profiles (Fig. 5D). Similar to our dataset, cluster A represented the most naïve and D represented the most mature Treg state (Fig. 5E, F). Cluster A, which showed high expression of naïve genes and *IL4R*, was overrepresented in Tregs from donors with T1D in comparison to healthy donors, whereas cluster D, which showed high expression of Treg effector genes, was overrepresented in healthy donors in all three datasets (Fig. 5G, H).

In the next step, we examined FOXP3+ Tregs in our cohort using flow cytometry (Supplementary Fig. 18A, B) and analogous data from the HPAP database (Supplementary Fig. 18C, D). In both cases, there were slightly more FOXP3+ cells among CD4+ T cells in children with T1D than in healthy donors. However, we did not see consistent differences in the expression of the naïve or effector signature genes between children with T1D and healthy donors. A possible reason is the low expression of *FOXP3* in the naïve Treg subset (Fig. 4C, D), which makes it difficult to identify these cells by the conventional flow cytometry panels, which highly depends on FOXP3 detection.

### Low frequencies of effector-phenotype unconventional T cells in children with T1D

It has been proposed that a loss of unconventional CD3+ CD56+ T cells (named TR3-56) with regulatory properties might contribute to the development of T1D[16]. In our cohort, we observed that children with T1D at T0, but not at T1, exhibited slightly lower expression of the TR3-56 signature than healthy donors, both in CD4+ and CD8+ T cells (Fig. 6A). We observed similar results in two unrelated cohorts with publicly available data (Supplementary Fig. 19A).

In the next step, we aimed to identify TR3-56 cells in our scRNAseq data, starting with the CD8+ T cells atlas, consisting of naïve and antigen-experienced conventional T, NK, γδT cells, and MAIT cells (Fig. 6B, C and Supplementary Fig. 19B). Among these populations, TR3-56 cells were predominantly annotated within the MAIT and γδ T cell subsets (Fig. 6B, C and Supplementary Fig. 19C). Accordingly, these clusters contained the most cells coexpressing CD3 and CD56 (Supplementary Fig. 19D). Furthermore, TR3-56 cells in these clusters exhibited the highest annotation scores (Fig. 6D), supporting their overlap with MAIT and γδ T cells.

To identify unique markers of TR3-56 cells independent of their parental subsets, we performed differential gene expression analysis comparing TR3-56 cells and non-TR3-56 cells in the CD8+ lymphocyte clusters (Fig. 6E). This analysis revealed that TR3-56 cells upregulate transcription factors associated with cytotoxicity and T cell activation (*RUNX3, BHLHE40, NR4A3*), as well as genes associated with homing to tissues (*CCR6, ITGB1*). Finally, we assessed the regulatory potential of TR3-56 cells by comparing their gene expression to a canonical Treg

signature, and found that TR3-56 cells exhibited the highest similarity among all CD8+ lymphocyte subsets. (Fig. 6F).

Re-clustering of the unconventional T cells identified two clusters of γδT cells and four clusters of MAIT cells (Fig. 6G, H). We observed that the γδT1 cluster was enriched in healthy donors in comparison to children with T1D at T0 and T1 (Fig. 6I). This cluster corresponded to effector-phenotype γδT cells (Fig. 6H), suggesting that the state of γδT cells in T1D essentially phenocopies conventional T cells (Fig. 2A–D). In contrast, γδT2 cells, enriched in some children with T1D (Fig. 6I), expressed lower levels of effector genes (Fig. 6H). Accordingly, the total unconventional CD8+ T cells from children with T1D express lower levels of cytotoxic effector genes such as *GZMB, GZMH, GNLY,* and *CCL5* (Supplementary Fig. 19E).

Using flow cytometry, we observed that CD8low T cells are enriched for γδTCR+ γδ56+ cells (Fig. 6J and Supplementary Fig. 20A). Thus, gating for CD8low cells can be used as a proxy for CD8+ γδT cells or CD8+ CD3+ CD56+ T cells. In our flow cytometry analysis, we found that percentages of γδT cells and CD8low T cells among CD3+ cells were low in children with T1D at T0, which was partially reverted at T1 (Fig. 6K and Supplementary Fig. 20A). Moreover, a higher percentage of CD8low T cells expressed *GZMB* in healthy donors than in children with T1D, essentially confirming the scRNAseq data (Fig. 6K). A higher percentage of CD8low T cells among CD3+ T cells from healthy donors compared to donors with T1D was observed also in the HPAP flow cytometry data (Supplementary Fig. 20B, C).

In the CD4+ T cell compartment, the Temra cluster contained most TR3-56 phenotype cells (Fig. 7A–C) and had the gene expression profile most similar to TR3-56 cells (Fig. 7D, E). Re-clustering of the Temra cluster revealed four subclusters (Fig. 7F). The Temra2 cluster, corresponding to the cells with the strongest cytotoxic phenotype and the highest expression of *GZMB* (Fig. 7G), was more abundant in healthy donors than in children with T1D at T0 (Fig. 7H).

### The analysis of T cell receptor repertoire in children with T1D

Alongside the single-cell transcriptomics, we analyzed T cell receptor sequences in T cells from the children with T1D and healthy donors. We observed only minimal clonal overlap between individual donors regardless of their disease status (Fig. 8A). However, there was a significant overlap of TCRα and TCRβ sequences in the same children with T1D at T0 and T1 (Fig. 8A), indicating the persistence of these clones in the blood during the one year period. We did not identify any T cell clones previously associated with T1D. The usage of variable, diversity, and joint segments did not significantly differ between the children with T1D and healthy donors (Supplementary Figs. 21, 22).

It has been reported previously that the CDR3 segments of TCRβ chains in people with T1D are generally shorter than in healthy individuals[73]. However, we did not observe any differences in the CDR3β length in CD4+ or CD8+ T cells in our cohort or in TCR repertoire profiling data from the HPAP database (Fig. 8B–D).

Finally, we analyzed the biochemical properties of the CDR3β TCR sequences in our cohort and in the HPAP cohort. We observed that the CDR3β sequences from people with T1D showed significantly higher Boman index and H Moment and a slightly lower Hydrophobicity index (Fig. 8E). This effect was apparent also on the average values per donor (Fig. 8F, G), but with a relatively low significance. A large cohort is required for eventual confirmation of this observation.

## Discussion

In this study, we profiled peripheral blood T cells from children newly diagnosed with T1D and healthy donors using scRNAseq and flow cytometry, with follow-up sampling one year after diagnosis to track longitudinal changes. One of our key findings was a reduced effector phenotype and upregulation of stemness-associated gene signature in T cells from children with T1D, evident in conventional CD4+ and CD8+ subsets as well as unconventional populations such as γδT cells.

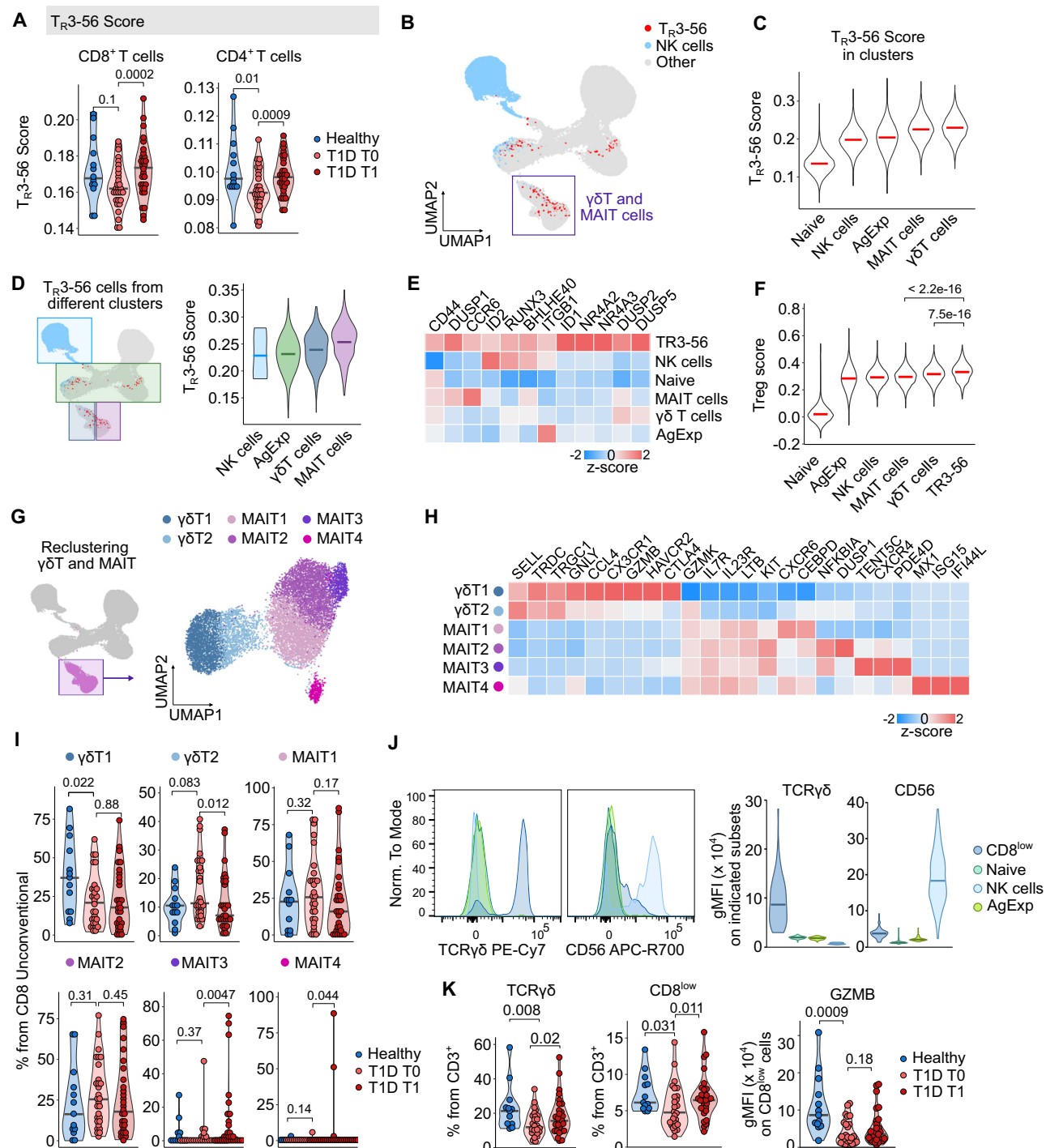

**Fig. 6 | Unconventional CD8⁺ T cells in children with T1D. A–C** Cells were annotated using a published RNA-seq dataset of FACS-sorted TR3-56 (CD3⁺CD56⁺), NK, CD3⁺CD56⁻, and CD8⁺ cells (GSE106082) with SingleR. **A** Quantification of TR3-56 annotation scores in CD8⁺ and CD4⁺ T cells from healthy donors (n = 13) and children with T1D at T0 (n = 30) or T1 (n = 29). **B** UMAP (from Supplementary Fig. 5A) with cells annotated as NK or TR3-56 highlighted. **C** Violin plots of TR3-56 annotation scores across clusters; bars at medians. **D** Quantification of TR3-56 annotation scores for cells from different parent clusters (only cells annotated as TR3-56 considered). UMAP highlights are illustrative; violin plots show medians. **E** Heatmap of marker genes shared across TR3-56 cells, column-scaled by z-score. **F** Module scores of Treg signature genes in main CD8⁺ clusters and TR3-56 cells. Violin plots; P-values by two-tailed Wilcoxon test. **G** Reclustering of CD8⁺ γδT and MAIT cells from Supplementary Fig. 19B (11,012 cells, 43 donors) after reprocessing.

UMAP colored by clusters. **H** Heatmap of marker genes defining clusters in (**G**). **I** Percentage of cells from clusters in (**G**) among total unconventional CD8⁺ T cells per sample in healthy (n = 13), T1D T0 (n = 30), and T1D T1 (n = 29). **J, K** Flow cytometry validation of CD8⁺ TR3-56 cells (gating in Supplementary Fig. 20). **J** Intensity of TCRγδ and CD56 in naïve CD8⁺, non-naïve CD8⁺, NK, and CD8low T cells. Shown: representative histogram (left) and quantification across 68 samples (n = 13 healthy, n = 26 T1D T0, n = 29 T1D T1). **K** Frequencies of γδT (left) and CD8low (middle) T cells among CD3⁺ cells, and GZMB intensity in CD8low cells (right), across the same donors. Panels (**A, I, K**): P-values by two-tailed Mann–Whitney test (healthy vs. T1D T0) or two-tailed paired Mann–Whitney test (T1D T0 vs. T1). Bars = medians. AgExp – Antigen Experienced T cells, MAIT – Mucosa Associated Invariant T cells, T1D – Type 1 Diabetes Mellitus, T0 – timepoint at diagnosis, T1 – timepoint one year after diagnosis.

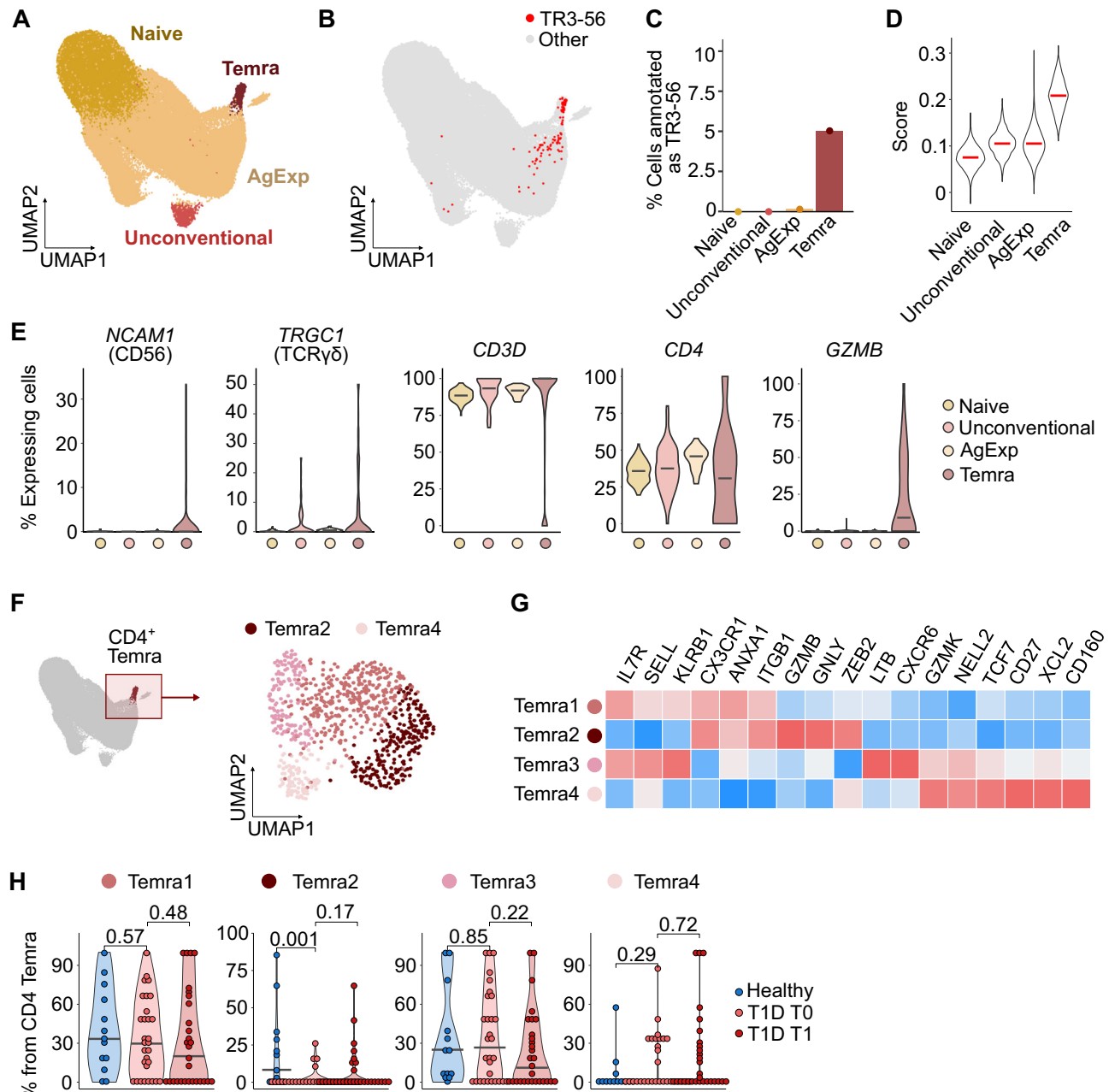

**Fig. 7 | Unconventional CD4⁺ T cells in children with T1D. A** UMAP of CD4⁺ T cells showing merged Louvain clusters used for subset analysis (n = 79,876 cells from 43 donors). **B** Same UMAP annotated with a published dataset of TR3-56 (CD3⁺CD56⁻), NK cells, CD3⁺CD56⁻, and CD8⁺ cells (GSE106082) using SingleR. Cells annotated as TR3-56 are highlighted. **C** Percentage of TR3-56–annotated cells in each cluster. **D** Violin plots of TR3-56 annotation scores (similarity to TR3-56 reference) by cluster; bars = medians. **E** Violin plots of the percentage of cells per cluster with non-zero expression of selected genes (87 samples); bars = medians. **F** Reclustering of CD4⁺ Temra cells from (**A**). In the resulting UMAP, there are n = 815 cells from 43 donors. **G** Heatmap of selected marker genes for clusters in (**F**), colored by row-scaled z-scores of average expression. **H** Percentage of cells in subclusters from (**F**) among total CD4⁺ Temra cells in healthy (n = 13), T1D T0 (n = 30), and T1D T1 (n = 29) donors. *P*-values by two-tailed Mann–Whitney test (healthy vs. T1D T0) or two-tailed paired Mann–Whitney test (T1D T0 vs. T1); bars = medians. AgExp – Antigen Experienced T cells, T1D – Type 1 Diabetes Mellitus, T0 – timepoint at diagnosis, T1 – timepoint one year after diagnosis.

Expression of cytotoxic genes such as *GZMB* was downregulated in T cells from children with T1D across various subsets, which is coun-terintuitive given that direct T cell cytotoxicity against pancreatic β-cells has been proposed as a T1D-inducing mechanism[74–76]. The down-regulation of the effector genes became less pronounced at one year after diagnosis, but still apparent compared to healthy donors. These findings were supported by analyses of HPAP spleen and blood data-sets and by reanalysis of multiple public RNAseq studies. Consistently, previously published flow cytometry analysis revealed higher fre-quencies of naïve T cells and lower frequencies of terminally

differentiated Temra cells in adults at the onset of T1D in comparison to healthy donors[18]. Experiments on non-obese diabetic (NOD) mice showed that stem-like pancreatic antigen-specific CD8⁺ T cells were more effective at disease initiation than their effector counterparts, suggesting that effector function may arise later, at the site of inflammation[77]. Recently, we observed a similar pattern in NOD mice, where animals that developed diabetes exhibited a naïve-like pheno-type, whereas those protected from disease showed stronger antigen-specific activation[78]. Accordingly, it has been shown that pancreatic β cell-specific CD8⁺ T cells retain a stemness-associated epigenetic

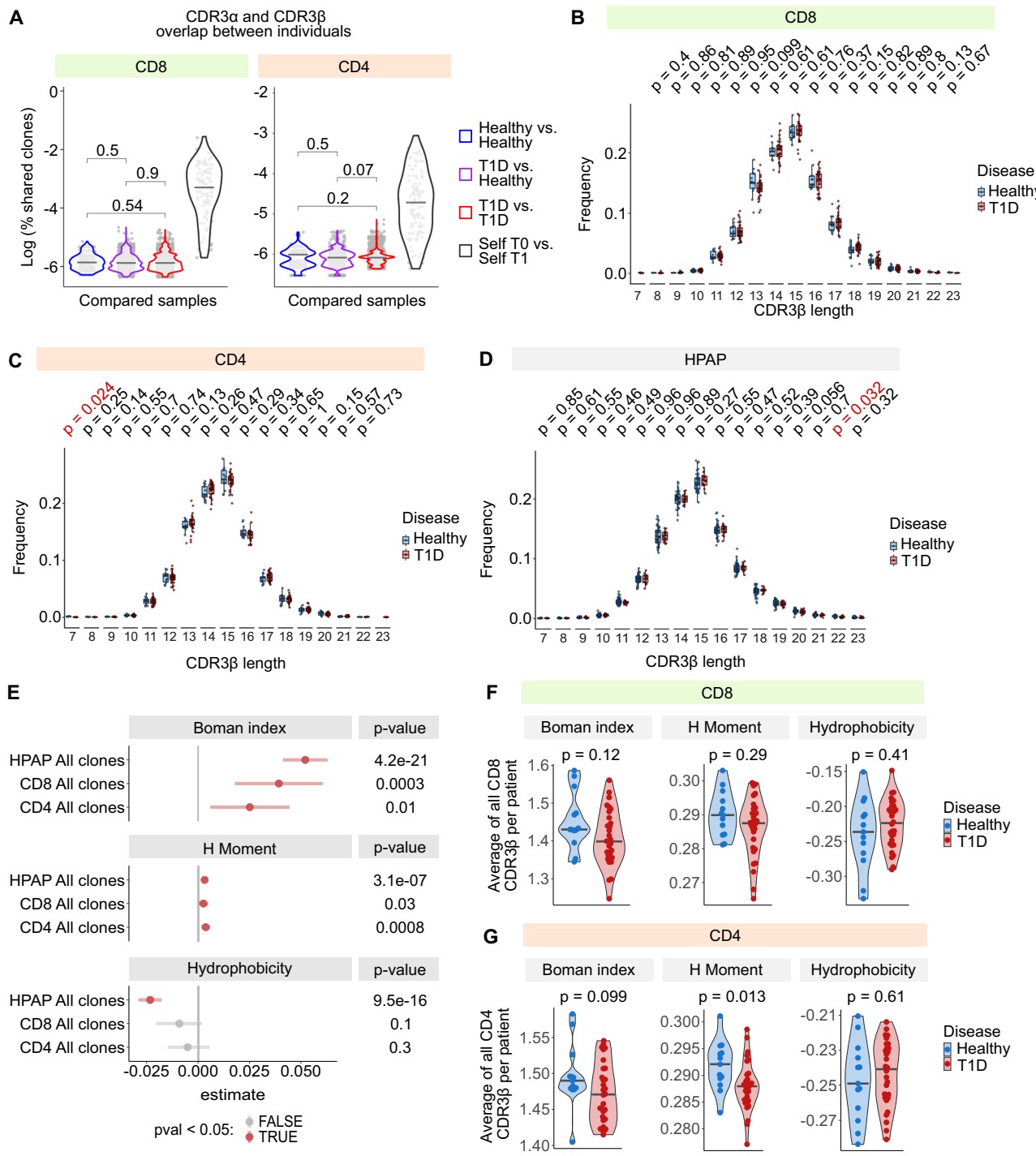

**Fig. 8 | TCR repertoires in children with T1D. A–C** TCR repertoires of CD4[+] and CD8[+] T cells from children with T1D at T0 (*n* = 30) or T1 (*n* = 29) and healthy donors (*n* = 13) were profiled by 10x immune profiling. **A** Unique CDR3α/β amino acid sequences were extracted and used for overlap analysis. For each donor, the percentage of overlapping sequences with all others was calculated. Overlaps are shown for healthy–healthy (blue), healthy–T1D (violet), and T1D–T1D (red), as well as self-overlap between T0 and T1 within the same child. Analyses were restricted to conventional CD8[+] (excluding MAIT, NK, γδT) or CD4[+] (excluding iNKT) T cells. *P*-values by two-tailed Mann–Whitney test. **B–D** Length distributions of unique CDR3β sequences per donor. Frequencies were quantified and visualized for: (**B**) CD8[+] conventional cells (*n* = 30 T1D, 13 healthy), (**C**) CD4[+] conventional cells (*n* = 30 T1D, 13 healthy), and (**D**) mixed CD4[+]/CD8[+] splenic T cells from HPAP (*n* = 16 T1D, 38 non-diabetic). *P*-values by two-tailed Mann–Whitney test without correction for

multiple comparisons. Boxplots show medians, IQRs, and 1.5 × IQR whiskers. **E–G** Biochemical properties of unique CDR3β sequences (Boman index, H moment, hydrophobicity) were computed with the Peptides R package. Analyses included conventional CD8[+] T cells (excluding MAIT, NK, γδT; *n* = 30 T1D, 13 healthy), conventional CD4[+] T cells (excluding iNKT; *n* = 30 T1D, 13 healthy), and splenic T cells from HPAP (*n* = 17 T1D, 39 non-diabetic). **E** Clonotypes pooled within groups; contrasts shown for T1D vs. healthy in CD4[+], CD8[+], and HPAP T cells. *P*-values by two-tailed Mann–Whitney test. Estimates represent the difference of location (median of differences between T1D and healthy CDR3β sequences), with 95% CIs. **F**, **G** Donor-level averages of biochemical scores. Differences between T1D and healthy donors shown for CD8[+] (**F**) and CD4[+] (**G**). *P*-values by two-tailed Mann–Whitney test; bars at medians. T1D – Type 1 Diabetes Mellitus, HPAP – Human Pancreas Analysis Program.

signature in the blood of T1D patients[79]. Our observation of reduced cytotoxic and enhanced stemness-associated gene signatures in peripheral T cells aligns with these insights and supports the concept of altered differentiation dynamics in early T1D.

The cause of this stem-like phenotype bias of the T cell compartment in the children with T1D is unclear. One possibility is that it is the consequence of the metabolic imbalance induced by the disease. However, we did not see particularly low expression of the effector genes in children with ketoacidosis, indicating a lack of correlation with the severity of the metabolic disorder. Another possible explanation is that the insulin deficiency directly affects the state of the T cell compartment via hypostimulation of the insulin receptor signaling pathway, which was previously shown to promote T cell effector function in mice in an intrinsic manner[80]. However, we observed reduced expression of the cytotoxic genes in children with T1D one year after their diagnosis, albeit less pronounced than upon diagnosis, and in adults with T1D from the HPAP database, who were already receiving exogenous insulin therapy. Thus, we find it more likely that the phenomenon of low effector and cytotoxic profile in T cells precedes or coincides with the onset of clinical T1D. This scenario is in line with the hygiene hypothesis, which explains the current rise of autoimmune diseases in high-income countries by a reduced exposure of individuals to infectious agents, leading to immaturity of the immune system, which results in the loss of self-tolerance[10,81,82]. The evidence for the hygiene hypothesis in T1D comes from experiments on NOD mice, which show higher incidence of diabetes in germ-free conditions than in standard housing facilities[83], and from population studies[84,85]. An alternative and seemingly contradictory hypothesis postulates that T1D is triggered by viruses, such as enteroviruses[7,8,86]. These two phenomena can be reconciled as it is possible that individuals with an immature immune system and/or those being infected at an older age with the hypothetical triggering virus might be at a particularly high risk of by-stander T1D development. This scenario goes along with the hypothesis that the immune system is shaped by the environment within a relatively narrow window during early life[87].

Tregs are crucial for maintaining peripheral tolerance. For this reason, the lack or dysfunction of Tregs has been studied as a potential cause in many autoimmune diseases[88]. In the T1D research, conflicting findings have been reported, showing reduced[18,89], normal[90–96], or elevated[19,97,98] FOXP3+ Treg frequencies in the peripheral blood in individuals with T1D compared to healthy donors. Our and HPAP flow cytometry data showed slightly elevated Tregs among CD4+ T cells in T1D. These discrepancies might be at least partially caused by differences in the age or time after T1D onset in the particular cohorts. Accordingly, it has been shown that the frequencies of Tregs increase within the first year after T1D diagnosis[99].

Partial dysfunction of Tregs and their impaired maturation in individuals with T1D have also been proposed[19,30,91,92,97,100–102]. Despite the controversies and open questions surrounding the role of Tregs in T1D, Treg-targeted therapies have been used to treat T1D in preclinical models[103] and are being tested in clinical trials in T1D patients[104–106]. In this study, we observed that a relatively high proportion of Tregs from children with T1D show low expression of key Treg effector genes such as IL2RA, FOXP3, CTLA4, TGFB1 and high expression of CD226, IL4R and GZMK, which are connected with Treg dysfunction[60,61,71,72] or T cell senescence[107]. Moreover, the whole Treg compartment was slightly altered in children with T1D, as it was enriched for the gene signature specific for dysfunctional FOXP3-deficient Treg-like cells, such as those present in IPEX patients or FOXP3 knock-out Tregs[66–70]. T1D is one of the typical autoimmune symptoms in IPEX[108]. In contrast to the pleiotropic IPEX syndrome, the children in our cohort developed only T1D and were still self-tolerant to most tissues, indicating that their Treg cells were still largely functional. However, their Treg compartment might be prone to failure in a specific context, such as suppressing an autoimmune response towards the pancreatic β-cells[109].

Accordingly, the progressive loss of a specific subset of effector KLRG1+ ICOS+ Tregs is associated with the T1D development in NOD mice[110]. We speculate that the bias of the Tregs in children with T1D towards the less effector phenotype and their similarities to genetically dysfunctional Tregs might contribute to T1D development.

A population of CD3+ and CD56+ co-expressing lymphocytes referred to as TR3-56 was proposed to be a regulatory subset preventing diabetes[16]. To date, this subset has not been characterized using single-cell transcriptomics. We did not see CD3 and CD56 co-expressing T cells as a standalone subset, but rather as a heterogeneous group of cells overlapping with unconventional subsets such as CD8low γδT and MAIT cells, and CD4+ Temra cells. We observed generally reduced frequencies of these subsets in children with T1D in comparison to healthy donors at the time of their diagnosis. Low frequencies of γδT in individuals with T1D upon diagnosis and one year later have been reported previously[111]. The same study showed a drop in CD56+ γδT cells at the time of T1D diagnosis, which was reverted after one year, which aligns with our data[111]. Moreover, these unconventional cells showed generally more naïve and less effector phenotype in T1D patients at both time points in comparison to healthy donors in our cohort. It is unclear if these unconventional T cells are involved in the self-tolerance and T1D prevention or if their less cytotoxic phenotype only corresponds to the overall state of the T cell compartment in children with T1D.

We used the data from our cohort to address previously proposed differences between people with T1D and healthy donors. Whereas a previous study observed elevated expression of butyrophillin-encoding genes BTN3A2 and BTN3A3 in T1D patients[26], we observed the opposite. Our analysis of data generated by multiple studies showed that the expression of BTN3A2, residing in the extended MHC region, depends on particular HLA haplotypes. Whereas our healthy donors were not HLA-matched to the people with T1D, the previous study used partially matched controls[26], which might explain the different observations.

Our analysis of the TCR repertoires revealed clonal expansion in the antigen-experienced subsets and showed that the same clones can be detected in the same children one year after analysis. However, we could not detect any public clones present in different donors. Our cohort, as well as data from the HPAP, did not show changes in the CDR3 length between people with T1D and healthy donors, which were observed previously[73]. We detected small differences in the average biophysical parameters of the TCRs between children with T1D and healthy donors, such as Boman index, H Moment, and hydrophobicity. However, additional studies are required to validate these findings.

Our analysis is the first comprehensive analysis of CD4+ and CD8+ T cells from newly diagnosed children with T1D and the same individuals at one year after the diagnosis using single-cell transcriptomics. It indicated differences between children with T1D and healthy donors at the time of diagnosis, some of which persist to the one-year time point post-diagnosis. Our findings indicate an overt stem-like phenotype in the T cell compartment, potential decreased functionality in the Treg compartment, and alterations in unconventional T cell subsets. These findings were largely supported by our flow cytometry analysis and/or analyses of data published by others, when possible. However, our experimental approach still needs to be taken as an exploratory study in a relatively small cohort. Further studies addressing these conclusions are needed.

## Methods
### Collection of clinical metadata
The samples were collected between February 2021 and February 2023 at a setting of a tertiary center for pediatric diabetes (Motol University Hospital, Prague, Czechia). After the consent of parents/caregivers and the study participants was granted, two study samples were obtained from each participant. The first sample was obtained at a median of 9

(4–24) days after the clinical diagnosis of T1D, the second sample at a median of 377 (359–423) days after diagnosis. At both time points, fasting C-peptide, HbA1c, and the standard blood count with the differential count were performed. At T0, the measurements of the T1D-specific autoantibodies (a-GAD, IAA, a-IA2, a-ZnT8) were performed. Data on diabetic ketoacidosis status at first contact (pH below 7.3 and/or bicarbonate < 15) were obtained from individual medical records. To assess partial clinical remission at T1, we employed the insulin dose adjusted HbA1c (IDAA1c) as a marker. IDAA1c was calculated as HbA1c (%) plus [4 times insulin dose (units per kilogram per 24 h)]. IDAA1c < 9 was considered as the presence of partial clinical remission[112].

The samples were obtained from 30 children with T1D at T0 and from 29 of these children at T1. One T1D participant withdrew from the study before the one-year follow-up visit. Their sample was included in all analysis except for paired statistical testing. A cohort of 13 healthy donors was also enrolled. Healthy donors were frequency-matched to the T1D cohort by age. Healthy donors were not HLA-matched to participants with T1D. The parents/caregivers and the participants signed a written consent.

## Processing of blood samples

Three to ten mL of peripheral blood were collected into EDTA-coated tubes, kept on ice, and transferred from Motol University Hospital to the Institute of Molecular Genetics within two hours. PBMCs were separated using Ficoll-Paque (GE Healthcare) and immediately frozen following a cryopreservation protocol (10x Genomics). Briefly, 4 ml of Ficoll-Paque was overlaid with blood and centrifuged at $400 \times g$ for 30 min at room temperature (brake set to one). The mononuclear cell layer was washed in PBS and resuspended in RPMI medium containing 40% FBS. After adding an equal volume of freezing medium (RPMI, 40% FBS, and 30% DMSO), two to five aliquots of PBMCs were frozen and stored in liquid nitrogen. PBMCs were gently thawed by slow, sequential dilution in RPMI medium containing 10% FBS. To minimize cell stress, wide-bore tips were used during the thawing process.

## ScRNAseq experimental design

ScRNAseq was performed in two different experiments, hereafter referred to as the initial experiment and the final experiment.

## Initial scRNAseq experiment

The initial experiment was performed in two batches: batch 1 – CD8$^+$ T cells from 6 children with T1D and 2 healthy donors in one well, batch 2 – CD4$^+$ T cells from 6 children with T1D and 3 healthy donors, CD8$^+$ T cells from 1 healthy donor in one well, and CD4$^+$ and CD8$^+$ cells from 6 children with T1D and 3 healthy donors in two wells. Cells were incubated on ice for 5 minutes with Human TrueStain FcX (BioLegend #422301) and for additional 30 min with anti-human CD4 and CD8 antibodies (CD8 APC, LT8, Exbio #1A-817-T100; CD4 AF700, MEM-241, Exbio # A7-539-T100) and one of the hashtag antibodies (TotalSeq™-C0251 anti-human Hashtag 1 Antibody, LNH-94 2M2, BioLegend #394661; TotalSeq™-C0252 anti-human Hashtag 2 Antibody, LNH-94 2M2, BioLegend #394663; TotalSeq™-C0253 anti-human Hashtag 3 Antibody, LNH-94 2M2, BioLegend #394665; TotalSeq™-C0254 anti-human Hashtag 4 Antibody, LNH-94 2M2, BioLegend #394667; Total-Seq™-C0255 anti-human Hashtag 5 Antibody, LNH-94 2M2, BioLegend #394669; TotalSeq™-C0256 anti-human Hashtag 6 Antibody, LNH-94 2M2, BioLegend #394671; TotalSeq™-C0257 anti-human Hashtag 7 Antibody, LNH-94 2M2, BioLegend #394673; TotalSeq™-C0258 anti-human Hashtag 8 Antibody, LNH-94 2M2, BioLegend #394675; Total-Seq™-C0259 anti-human Hashtag 9 Antibody, LNH-94 2M2, BioLegend #394677; TotalSeq™-C0260 anti-human Hashtag 10 Antibody, LNH-94 2M2, BioLegend #394679) and with Hoechst 33258 for viability right before sorting. From each sample, 10,000 CD4$^+$ cells and 10,000 CD8$^+$ cells were sorted. All samples were collected to the same collection tube, washed with PBS/0.05% BSA and counted using the TC20

Automated Cell Counter (#1450102, Bio-Rad). The viability of the cells before loading was higher than 85%.

## Final scRNAseq experiment

The final experiment was performed in four batches: batch 1 – CD4$^+$ and CD8$^+$ T cells from 8 children with T1D T0, 8 children with T1D T1, 4 healthy donors in 4 wells, batch 2 - CD4$^+$ and CD8$^+$ T cells from 8 children with T1D T0, 8 children with T1D T1, 4 healthy donors in 4 wells, batch 3 – 8 children with T1D T0, 7 children with T1D T1, 3 healthy donors in 4 wells, batch 4 – 6 children with T1D T0, 6 children with T1D T1, 4 controls in 4 wells. Cells from the same donor from time 0 and time 1 were always processed in the same well to minimize batch effect. Cells were incubated on ice for 5 minutes with Human TrueStain FcX (BioLegend #422301) and for additional 30 minutes with anti-human CD4, CD8, CD45RA and CCR7 antibodies (CD4 APC, MEM-241, Exbio #1A-359-T100; CD8 PE, MEM-31, Exbio #1P-207-T025; CD45RA FITC, MEM-56, Exbio #1F-223-T100; CCR7 PeCy7, G043H7, BioLegend #353226) and one of the hashtag antibodies (the same as in the initial experiment) and with Hoechst 33258 for viability right before sorting. Non-naïve cells were enriched in the final sample as follows: from each sample 1000 naïve (CCR7$^+$ CD45RA$^+$) and 5000 antigen-experienced cells were sorted into two tubes, one for CD4$^+$ T cells and one for CD8$^+$ T cells. All samples were collected to the same two collection tubes, washed with PBS/0.05% BSA and counted using the TC20 Automated Cell Counter (#1450102, Bio-Rad). The viability of the cells before loading was higher than 85%. One control sample was removed from the analysis because of the bad quality of the data at the sort.

Cells from both cohorts were loaded onto a 10x Chromium machine (10x Genomics) aiming for a yield of 1500 cells per sample. cDNA libraries were prepared using the Feature Barcode technology for Cell Surface Protein protocol (#CG000186 Rev D) with the Chromium Single Cell 5′ Library & Gel Bead and Chromium Single Cell 5′ Feature Barcode Library kits (10x Genomics, #PN-1000014, #PN-1000020, #PN-1000080, #PN-1000009, #PN-1000084) according to the manufacturer's instructions. Sequencing was performed on the NovaSeq 6000 platform (Illumina), yielding an average of 45,745 reads per cell in the gene expression libraries and 3712 reads per cell in the V(D)J libraries (Supplementary Table 4).

## Quality control, normalization, and integration of scRNAseq data

The human reference used to map sequenced reads was taken from Ensembl version 102[113] and prepared using 10x Cell Ranger 5.0.1 Software (*mkref* tool). The count matrices were generated by 10x Cell Ranger 5.0.1 Software (*count* tool) in either R2-only or paired-end mode. Afterwards, they were pre-processed using the Seurat 4.3.1 package[114] on R 4.2.1. Any cell that was not marked by any expected combination of hashtags was removed. All cells with more than 10% of genes mapping to mitochondrial genes, those expressing less than 200 genes and those that were marked as doubles according to the V(D)J (more than 2 TCRα or more than 1 productive TCRβ sequence present) were excluded during initial object creation (code available in GitHub folder code01_raw_to_init).

Mitochondrial genes, ribosomal genes, genes encoding TCR variable segments (any gene symbol containing the TR[AB][VDJ] substring) and genes present in less than 3 cells were removed. Each data set was then normalized (default method and scale factor = $1 \times 10^4$), scaled, subjected to dimensional reduction (PCA with 20 principal components followed by UMAP) and Louvain clustering. PTPRC counts were generated by the IDEIS tool and normalized by the centered-log ratio method[33].

Preprocessed datasets were merged, normalized, scaled and used for dimensional reduction (PCA with 12 principal components and UMAP) and Louvain clustering. In a second, more stringent quality control step performed on the merged dataset (code available in

GitHub folder code02_analysis_of_init, parts 02 and 03), clusters of dead or low-quality cells and individual cells with fewer than 500 detected genes were excluded. Integration of datasets from different batches was performed using STACAS (v 2.1.3)[115]. In a subclustering step for CD4[+] Temra cells, we revealed a contaminating cluster of NK cells ($n = 80$), which only appeared after reclustering. We kept those cells in the final dataset, but removed them from the analysis in Fig. 7.

## Analysis and visualization of scRNAseq data
Transcriptional regulatory interactions were estimated using the decoupleR package (v2.10.0) using the CollecTRI database[116]. GSEA pathways were processed using gene sets from datasets GSE9650[41] and GSE11057[42] with the R package fgsea (v1.20.0)[117]. Heatmaps were created with the package pheatmap (v 1.0.12). Sankey plots for Supplementary Fig. S3 and S4 were created using RAWGraphs (v2.0)[118].

DEGs were identified using the two-sided Wilcoxon rank-sum test (Seurat FindMarkers), with p-values adjusted for multiple testing using the Bonferroni correction. To address concerns about pseudoreplication in cell-level differential expression analysis, we performed a sample-level pseudobulk analysis. For each annotated T cell subset, raw gene expression counts were aggregated per donor, generating one pseudobulk profile per donor per subset. Differential gene expression and log fold changes were then calculated using the DESeq2 package (v1.44.0), enabling rigorous comparison between healthy donors and children with T1D at different timepoints. In parallel, we applied the decoupleR package (v2.10.0) to quantify the similarity of each donor's expression profile to reference gene expression contrasts between naïve and memory T cells (GSE179613).

Pseudotime analysis was performed using the slingshot package (v2.12.0) to infer a lineage trajectory among Treg subsets. The analysis was based on UMAP embeddings and cluster annotations. To test for differences in pseudotime progression between groups, we used the condiments package (v1.12.0), which estimates group-specific changes in pseudotime distributions along the inferred trajectory.

## HLA Typing and allele frequency analysis
We used the raw fastq files for estimation of the HLA genotype using arcasHLA[119]. In cases where samples were profiled twice for the initial and final dataset and/or for two timepoints, fastq files were merged to one fastq file per donor. These fastq files were processed with the commands *genotype* and *merge*. Reference data from the global population of the Czech Republic were obtained from The Allele Frequency Net Database[55] using the tool HLAfreq[120].

## Analysis of published data
Count matrices from the RNA sequencing data were obtained from the Gene Expression Omnibus (GEO) (GSE237218[47], GSE123658 (unpublished), GSE10586[46]) and processed with DESeq2[121]. Raw fastq files from the Single Cell 3' sequencing of PBMC of four Finnish children at risk of developing Type 1 diabetes and their gender, age and HLA matched control children were obtained from the European Genome-Phenome Archive (EGAD00001005768[26]) and mapped with CellRanger software (10x Genomics, cellranger-7.1.0) to the *GRCh38* human reference genome. Count matrices were processed with Seurat similarly to our data. Processed scRNAseq data of PBMCs from 46 T1DM cases and 31 matched controls was obtained from Synapse under the accession code syn53641849[29]. Processed scRNAseq data of PBMCs from 5 T1M donors and 3 healthy donors was obtained from GEO GSE221297[30]. Processed scRNAseq data of PBMCs from 12 T1D donors and 12 healthy donors was obtained from ParseBio (Supplemental Table 1). Genes from all datasets were ranked by the fold changes obtained from comparing children with T1D to healthy donors (DESeq LogFC). For comparison between datasets, the ranks were converted to 20-quantiles.

For the genotype-expression analysis of BTN3A2, we used bulk RNA sequencing data from GSE237218[47], GSE123658 (unpublished),

EGAD00001005767 (Kallionpaa), and scRNAseq data from the current study and from the HPAP repository. In all cases, raw fastq files were processed using arcasHLA as described above to obtain HLA genotypes for all the donors. HLA alleles were typed to the level of two fields, which distinguishes specific HLA proteins. The information about the expression of BTN3A2 for each of the donors was obtained from the count matrices that were generated for each dataset as described above. The expression of BTN3A2 for the donors in each dataset was scaled to a range from −1 to 1 to allow comparison between datasets, i.e., the donor with the lowest expression of BTN3A2 in each dataset had a value of −1, and the donor with the highest expression in each dataset had a value of 1. The effect of study, diabetes status and particular alleles in all MHC-I and MHC-II loci was calculated using a generalized linear model with gaussian distribution. The analysis was performed for each locus separately.

## Annotation of cells with previously published signatures
For annotation of cells with the TR3-56 signature, count matrix from RNA sequencing of the flow sorted CD3[+] 56[+] cells, NK cells, CD3[+]CD56[-], and CD8[+] cells was obtained from the Gene Expression Omnibus (GSE106082)[16]. CD4 and CD8 datasets were annotated using SingleR[122].

For annotation of Treg cells, the differentially expressed genes from dataset GSE15659[58] were retrieved from the mSigDB (#M3563). The function AddModuleScore from the Seurat package was used to calculate the expression of the genes in the Treg dataset.

## Analysis of scRNAseq data from the HPAP database
From the HPAP repository, we used a collection of scRNAseq samples from splenocytes and lymph nodes of the deceased healthy or T1D donors that were processed using the HPAP CITEseq: Dual-index 3' HT scRNAseq with Antibody Derived Tags and Hashtag Oligos protocol. The full protocol can be accessed at the HPAP database. For our analysis, raw fastq files were downloaded from the HPAP web and mapped to the GRCh38-2020-A human reference genome using cellranger-7.1.0. Count matrices were merged and subjected to normalization (scale factor $= 1 \times 10^4$), scaling and dimensional reduction (PCA with 15 principal components on 2000 variable features and UMAP) and Louvain clustering. Clusters representing dead cells or contaminating cell types other than T or NK cells were removed. For analysis of the Treg subpopulations, splenic Tregs were extracted from the whole dataset and subjected to new to normalization (scale factor $= 1 \times 10^4$), scaling and dimensional reduction (PCA with 10 principal components on 1000 variable features and UMAP) and Louvain clustering. The batch effect of cells from different experiments was removed using STACAS (v 2.1.3).

## Analysis of TCR repertoires
TCR repertoires of the CD4[+] and CD8[+] T cells from initial and final experiments were profiled using 10x Chromium Single Cell V(D)J Enrichment Kit, Human T Cell (#PN-1000005) and mapped by the 10x Cellranger 5.0.1 sotware (*vdj* tool) to human reference obtained the International ImMunoGeneTics Information System (IMGT)[123] for the immune receptor repertoire profiling. Additional V(D)J sequences were extracted from the gene expression library using the MiXCR 3.0.12 software[124]. For the clonal expansion analyses, cells with the same nucleotide sequences of the CDR3α and CDR3β were considered clones. In cases where we detected two productive rearrangements of CDR3α, the cells were considered the same clones if the nucleotide sequences of CDR3β and both CDR3α were shared, or if the nucleotide sequences of CDR3β and one CDR3α were shared while the second CDR3α was not detected.

For the repertoire overlap analysis, unique amino-acid sequences of the CDR3α or CDR3β were extracted for each donor. Then, the percentage of overlapping sequences in one participant with all of the other participants was calculated and quantified in the following

comparisons: (i) healthy donors with other healthy donors, (ii) healthy donor with a T1D donor, or iii) two T1D donors. In addition, the Self T1 – Self T0 comparisons were calculated as the overlap between CDR3α or CDR3β sequences in the same donor in the two timepoints.

For the analysis of the length of the CDR3β, unique amino-acid sequences of the CDR3β were extracted for each donor. The lengths of the sequences were quantified as the frequency of each length in a particular donor and visualized in children with T1D and healthy donors.

For analysis of the biochemical properties of the CDR3 sequences, we used the R package Peptides[125]. For each of the biochemical properties available in the package, the values for all the unique CDR3β sequences were calculated and compared between healthy donors and children with T1D. Each sequence was taken into the analysis, the number of times it occurred in the dataset, i.e., a CDR3β sequence representing an expanded clone that was detected in three donors, with 50 occurrences in donor 1, 10 occurrences in donor 2 and 1 occurrence in donor 3 would be counted 61 times. To see the global biases of the repertoires in children with T1D and healthy donors not influenced by the clonal expansion, we calculated the biochemical properties for all the unique CDR3β sequences in each donor, and averaged these values per donor.

For all the analyses except for clonal expansion, the analysis was performed on conventional cells only, i.e., MAIT cells, NK cells, and γδT T cells were excluded from the CD8+ dataset, and iNKT cells were excluded from the CD4+ dataset.

The analyses of CDR3β length and biochemical properties were performed also on samples from splenocytes of the non-diabetic or T1D donors from the HPAP dataset. In the HPAP repository, the TCR libraries were prepared as described previously[126]. Briefly, T cell receptor beta chain gene rearrangements were bulk sequenced from gDNA using custom primers. Typically, 100 ng of input DNA was amplified per replicate. Sequencing libraries were prepared using 2 × 300 bp paired-end kits (Illumina MiSeq Reagent Kit v3, 600-cycle, Illumina Inc., San Diego, CA). The raw sequencing data (fastq files) were downloaded from the HPAP database and mapped to the using Mixcr v4.5.0 to the human reference obtained from the International ImMunoGeneTics Information System (IMGT)[123].

## Flow cytometry analysis

The list of used antibodies is provided in Supplementary Table 6. Flow cytometry was performed in two batches. In each batch, aliquoted PBMCs stored in the liquid nitrogen were gently thawed by slow, sequential dilution in RPMI medium containing 10% FBS. Cells were counted, and a comparable number of cells (1.2 million in batch one, 2 million in batch two, depending on sample availability) were used for staining. The counted cells were divided into three equal parts. The first part was used for extracellular staining of live cells, the other two parts were used for intracellular staining of fixed cells. To prevent unspecific binding to Fc receptors, Human TrueStain FcX (BioLegend #422301) was added to all staining mixes. For staining of extracellular markers (Panel 1: CD4 OKT4 BV421, CD16 BV510, TCR Vα24-Jα18 BV605, CD45RO BV650, CD197 BV711, CD45RA FITC, CD3 UCHT1 PE-Cy5, HLA-DR PerCP/Cy5.5, CD183 PE, CD25 PE/Fire700, CD196 PE/Dazzle 594, TCR gamma/delta PE-Cy7, CD8 APC, CD56 APC-R700), cells were incubated with diluted antibodies and LIVE/DEAD Fixable Near-IR Dead Cell Stain Kit (Invitrogen #L34975) for 30 min on ice immediately after isolation. For staining of intracellular markers and transcription factors, cells were first stained for 30 min on ice with the mix containing LIVE/DEAD Fixable Near-IR Dead Cell Stain Kit (Invitrogen #L34975) and extracellular markers (CD3 MEM-57 AF700, CD4 SK3 BV421, CD8 APC, CD25 PE/Fire700, CD45RA FITC, CD45RO BV650, CD45R/B220 BV510) and then fixed and permeabilized using the eBioscience Foxp3 / Transcription Factor Staining Buffer Set (Invitrogen #00-5523-00) according to the manufacturer's instructions,

washed, and stained overnight with antibodies for intracellular markers in 4 °C (Panel 2: extracellular markers and CD127 BV605, Ki-67 BV750, HLA-DR PerCP/Cy5.5, CD226 PE, FOXP3 PE/Dazzle 594, CD137 Pe/Cy7; Panel 3: extracellular marekrs and CD127 BV605, Ki-67 BV750, IκBα AF488, GZMK PerCP/Cy5.5, CD184 PE, Eomes PE/CF594, GZMB PE/Cy7).

Flow cytometry was carried out with a Cytek Aurora flow cytometer, configuration 4 L 16V-14B-10YG-8R (Cytek). Data were analyzed using FlowJo software v10.10 (BD Life Sciences). Anomalies in flow rate, signal acquisition, and dynamic range were removed using the FlowJo plugin FlowAI[127].

## HPAP flow cytometry

Flow cytometry data (fcs files) from PBMC of donors with T1D and healthy donors were downloaded from the HPAP. These samples were collected from deceased donors on the date of death and analyzed either fresh or after cryopreservation. For our analysis, we used only PBMC samples from donors with T1D and healthy donors. Two samples were excluded due to suspected duplicity, and six samples were excluded based on the lack of compensation metadata in the fcs file. The list of used and excluded samples is provided in Supplementary Table 7. Samples were analyzed in groups based on three different panels for staining and manually compensated when needed. We analyzed 41 samples in Panel 1 – CD4 phenotyping panel, 19 samples in Panel 2 – CD8 phenotyping panel and 26 samples in Panel 3 – CD8 phenotyping panel, focused primarily on antigen-specific cells. In cases when the same patient had measurements in multiple panels, these values were averaged. For analysis, we excluded donors youger than 5 years as appropriate age-matched heatlhy controls were not available.

## Statistical analysis

The statistical analysis was performed using tests indicated in the Figure legends using R v4.2.1. Quantitative scRNAseq and flow cytometry data were tested using the nonparametric Mann–Whitney test without a correction for multiple comparisons or, for a comparison of the same donors at two different timepoints, using the paired Wilcoxon signed rank test without a correction for multiple comparisons. Two-tailed tests were performed. The number of biological replicates (cells and donors) is indicated in the respective Figure legends.

## Bayesian statistics

Differences in population composition were assessed by a Bayesian generalized linear model with a negative binomial response, with an offset term to normalize for the total number of cells in a sample. The input data consisted the counts of cells for each Level 3 population, and the model included a fixed effect for the type of sample (healthy, T1D at time 1, T1D at time 0) and the Level 3 population. Additionally, random effects of the type of sample were added, grouped by both Level 2 and Level 3 populations. The shape parameter was also predicted, with a fixed effect for Level 2 populations and a random intercept for Level 3 populations. The model was fitted with the brms package[128]. We used the default priors in the brms package, specifically: (i) The intercept for the mean has student-t prior with 3 degrees of freedom and scale 2.5, with location guessed by brms from the overall range of the data (1.81 for CD4 data and 1.97 for CD8 data), (ii) Student-t prior with 3 degrees of freedom and scale 2.5 centered at 0 for the intercept for overdispersion, (iii) Improper flat prior for all fixed effects, (iv) Half Student-t prior with 3 degrees of freedom, scale 2.5 and location 0 for standard deviation of random effects, v) LKJ(1) prior for correlations between random effects.

## Ethical consent

The study was approved by the institutional Ethics Committee (EK-819/20) of the Motol University Hospital. Written informed consent was obtained from all the participants and their legal guardians.

**Reporting summary**

Further information on research design is available in the Nature Portfolio Reporting Summary linked to this article.

## Data availability

The raw sequencing data are protected and are not available due to data privacy laws. The processed scRNAseq data and flow cytometry data generated in this study have been deposited on Zenodo under https://doi.org/10.5281/zenodo.17280189 [https://doi.org/10.5281/zenodo.17280189] and in the NCBI Gene Expression Omnibus (GEO) under the accession code GSE309970. Data from previously published studies used for validation of our findings are available in the Gene Expression Omnibus (GEO) database under the following accession codes: GSE237218, GSE123658, GSE10586, GSE221297, in the European Genome-phenome Archive (EGA) under the following accession codes: EGAD00001005767, EGAD00001005768, and in the Synapse database under the accession code syn53641849 [https://www.synapse.org/Synapse:syn53641849]. All display items presented in the main manuscript and supplementary information can be reproduced from data and code that are available in public repositories. The raw numbers for charts and graphs are available in the Source Data file whenever possible. Source data are provided in this paper.. Data from previously published studies used for validation of our findings are available in the Gene Expression Omnibus (GEO) database under the following accession codes: GSE237218, GSE123658, GSE10586, GSE221297, in the European Genome-phenome Archive (EGA) under the following accession codes: EGAD00001005767, EGAD00001005768, and in the Synapse database under the accession code syn53641849 [https://www.synapse.org/Synapse:syn53641849]. All display items presented in the main manuscript and supplementary information can be reproduced from data and code that are available in public repositories. The raw numbers for charts and graphs are available in the Source Data file whenever possible. Source data are provided in this paper.

## Code availability

The code can be accessed at https://github.com/Lab-of-Adaptive-Immunity/dia. The code for the Bayesian analysis can be accessed at https://github.com/martinmodrak/diabetes_populace.

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

## Acknowledgements

We gratefully acknowledge Ladislav Cupak for technical assistance, Zdenek Cimburek and Matyas Sima (flow cytometry facility, IMG) for cell sorting, Sarka Kocourkova and Michal Kolar (Laboratory of Genomics and Bioinformatics, IMG) for cDNA library preparations, and Stepanka Pruhova, Stanislava Kolouskova, Barbora Obermannova, Lenka Drnkova and Lukas Plachy, who cared for the children with T1D during the course of the study, and Jachym Antonin Harwood and Jan Vecek for advice on the manuscript. VNi and BC are students of the Faculty of Science, Charles University in Prague. This project was supported by the Czech Science Foundation (22-21356S to OS), project National Institute of Virology and Bacteriology (Program EXCELES, LX22NPO5103 to OS)—funded by the European Union—Next Generation EU, Charles University Grant Agency (404222 to VN), European Union's Horizon 2020 research and innovation program under grant agreement No. 802878 (ERC Starting Grant FunDiT to OS), core funding provided by the Institute of Molecular Genetics of the Czech Academy of Sciences (RVO 68378050), and core funding provided by the Institute of Molecular Genetics of the Czech Academy of Sciences (RVO 68378050), and Czech Ministry of Health (conceptual support project to research organization 00064203 – FN Motol). Computational resources were provided by the e-INFRA CZ project (ID:90254), supported by the Ministry of Education, Youth and Sports of the Czech Republic. This manuscript used data acquired from the Human Pancreas Analysis Program (HPAP-RRID:SCR_016202) Database (https://hpap.pmacs.upenn.edu/), a Human Islet Research Network (RRID:SCR_014393) consortium (UC4-DK-112217, U01-DK-123594, UC4-DK-112232, and U01-DK-123716).

## Author contributions

V.N.e and Z.S. enrolled human donors and provided biological samples and metadata, A.N. processed and cryopreserved the biological samples and prepared cells for scRNAseq experiments, V.N.i and J.M. analyzed transcriptomic and TCR profiling data generated in this study, V.N.i analyzed publicly available transcriptomic data, V.N.i and B.C. performed and analyzed flow cytometry experiments, B.C. analyzed HPAP flow cytometry data, M.M. performed Bayesian statistical analysis, Z.S. and O.S. supervised the project and provided administrative tasks required for the study, V.N.i and O.S. wrote the manuscript. All authors revised the manuscript.

## Competing interests

The authors declare no competing interests.
