## [Transparent Peer Review file · Nature Communications]

Imbalance of stem-like and effector T-cell states in early type 1 diabetes across conventional and regulatory subsets

Corresponding Author: Dr Ondrej Stepanek

Version 0:

Reviewer comments:

Reviewer #1

(Remarks to the Author)

It is important to profile T cells in type 1 diabetes (T1D) since T1D is thought as an autoimmune disease. The study by Veronika et al. investigated the T cell subsets and their features in T1D by comparing with healthy donors, revealing a disrupted balance in T cell phenotypes across several T cell subsets in T1D patients. While the topic is intriguing, the study's persuasiveness is somewhat diminished by the presentation of the results. The authors should conduct more robust experiments or perform a more rigorous analysis to substantiate their findings.

Major concerns:

1. The authors performed batch correction on six batches from initial scRNAseq dataset and final scRNAeq dataset. However, there seems to be a significant batch effect between the initial and final datasets after batch effect correction (Figure S2J), and the batch effect still could significantly bias the results. It is much better to project all the cells on the UMAP plot to examine the integrated results, which could help we understand whether the T cell subsets are common or sample specific.
2. The authors examined the fraction of T cell subsets in healthy donor and T1D patients and their change mainly using FACS sorting (Fig 2D, 2F, 3E-F, 4F-H, 5H, 6F-G). However, the authors did not examine the fraction of T cell subsets in scRNAseq data. Whether the FACS-results and scRNAseq results are consistent?
3. The authors identified DEGs and signals of CD8+T cells (or CD4+T cells) between T1D patients and healthy donors without considering T cell subsets (Fig 2A). The author further identified the DEGs of each T cell subset between T1D patients and healthy donors (Figure S6B). However, it is still unclear whether the identified DEGs of these T-cell subsets are consistent with each other or T cell subset specific? The authors could make a more explicit figure to clearly show DEGs and signals are shared by multiple T cell subsets or specific T cell subsets.
4. The feature, repeatability and functions of these Treg subsets in T1D are not well demonstrated. E.g. the authors claim that Treg1 has phenotypes of naïve Treg. However, Treg1 also highly expressed IL4R, IL7R, LAG3, GZMK and CD226 (Fig 3B) which are memory T or effector T specific genes. How about the fraction of Treg1 in each sample based on scRNAseq data? Whether Treg1 are present in each sample? Is Treg1 more likely to differentiate into Treg4 in healthy donors than in T1D using trajectory analysis?
5. It seems that TR3-56 cells are also located on conventional CD8+ T cells (Fig4B). What are the differences of the TR3-56 cells on conventional CD8+ T cells and MAIT/γδ T?

Minor concerns:

1. The expression of effector genes (NKG7, GNLY, CCL4, CCL5, CX3CR1, TNFRSF4, TNF, and TNFRSF9) in EGAD00001005768 are different the other data. The authors should examine and discuss this inconsistency.
2. Contaminating NK cluster (Fig. 5F) should be removed at the quality control or cell annotation rather than the final analyses.
3. Some errors should be checked and corrected, such as
 - 1) The term "NK T cells" should be clarified. Is it referring to NK cells or Natural Killer T (NKT) cells? (Figures S3A-D)
 - 2) The expression of CD79A, a B cell marker, by iNKT cells is puzzling and requires an explanation.
 - 3) Legends should be added to figures for clarity (e.g., Figs 6B-6C, S9A, F.).

(Remarks on code availability)

It is OK

Reviewer #2

(Remarks to the Author)

(Remarks on code availability)

Reviewer #3

(Remarks to the Author)

The manuscript by Niederlova et al describes the analysis of T-cell population from children with and without T1D at two time points a year apart. The manuscript points to some novel aspects of human T cell biology relating to T1D. Several parts of the manuscript could be improved. These are listed below.

Excerpts from the text are in italics, my comments are in regular text

Major points:

1. The title not clear. I assume it is the cells, not the phenotypes, that are in imbalance.
2. To compare the T-cell compartments in healthy donors and T1D children... It is more respectful of people with T1D to refer to people, or children in this case, with and without T1D.
3. The composition and the matching of the individuals in the cohorts is critical to this study. 30 T1D vs 13 HC, but not HLA matched? What about age and sex?. This is shown diagrammatically in Figure1, but needs to be stated explicitly and clearly in the Results section.
4. First, we performed an initial scRNAseq analysis of CD4+ and CD8+ lymphocytes from 16 samples...Were these children with, or without T1D? How were these 16 samples chosen? Did the CD45RO/RA+ correlate with the subject's age?
5. Can the authors validate that impact of batch specific effects did not impact their analysis and interpretation of the data? Supplementary materials show plots, but can this reduction in batch effects be quantified more objectively?
6. Have the authors stratified for the age of the participants? Could the differences described be attributable to the age of the participants more than their diagnosis with, or without, T1D?
7. The first paragraph of the discussion is irrelevant and should be deleted. If the authors feel this information should be included it should be in the Introduction.

Minor points

1. In my opinion it is not true to say that there has been a 'steep increase in T1D incidence...' (Introduction paragraph 1). This should be reworded to something less hyperbolic.
2. (Introduction paragraph 2) Despite numerous studies in the field...please cite some of these studies
3. The Network for Pancreatic Organ Donors with Diabetes. Please include the acronym nPOD.
4. We used a balanced medium-sized cohort...it is not clear what 'balanced' refers to here. Medium-sized is also subjective, state the number of individuals in each cohort.
5. Blood of 30 donors with T1D was taken 4-9 (median 9) days...this is not possible. If the range is 4-9, the median cannot be 9.
6. The T1D patients showed higher overall blood lymphocytes and lower neutrophils at T0 in comparison to T1 and healthy donors (Fig. S1C). Give the data and the statistics in the text.
7. We compared the proportions of particular subsets using both frequentists...please correct typo.
8. ...which could be further divided to naïve, central memory (T_{cm}), T_{reg}, Th1/Th17, Th2, Temra, ISAGhi. Please define ISAGhi and Tsig in the text.
9. How many participants were included in the HPAP study which the authors reanalyzed?
10. First, we observed that our healthy donors generally followed the common distribution of HLA alleles in the Czech population 45 with the exception of overrepresented HLA-A*02:01, whereas most T1D children carried the respective susceptible alleles. This sentence is very unclear. The non-T1D subjects were more frequently A2+ than expected?...Even though there were only 13? What is meant by 'respective susceptibility alleles- HLA-DR4-DQ8, DR3-DQ2?

(Remarks on code availability)

Reviewer #4

(Remarks to the Author)

- What are the noteworthy results?

Niederlova et al have produced an interesting piece of research examining differences in T cell populations between children without T1D, children newly diagnosed, and those one year after diagnosis. They describe in great detail changes in effector T cell populations. A strength of the work is confirmation of the findings by re-examination of previously published scRNAseq and flow cytometry data sets.

- Will the work be of significance to the field and related fields? How does it compare to the established literature? If the work is not original, please provide relevant references.

This work is useful to the field as we attempt to understand phenotypes of both autoantigen- specific and non specific T cells in T1D. Other papers have performed scRNAseq and flow cytometric analysis of T cells before (and these are well described in the introduction of this manuscript), but the focus on children at point of diagnosis and after one year adds value in this study.

- Does the work support the conclusions and claims, or is additional evidence needed?
In general yes, but see major comments below on additional analysis required

- Are there any flaws in the data analysis, interpretation and conclusions? - Do these prohibit publication or require revision?
Yes please see the comments below that require revision

- Is the methodology sound? Does the work meet the expected standards in your field?
Yes

- Is there enough detail provided in the methods for the work to be reproduced?
For the wet lab work- yes.
For the data analysis -yes, but see comments on code

Major comments

"Processed data required to generate figures for the manuscript are available on Zenodo: DOI: 10.5281/zenodo.14222418 Raw data were not deposited to protect the identity of study participants." Under FAIR data principles, I would expect to have access to the raw data upon point of publication e.g. fastq files as much information is lost by running the cellranger pipeline and conversion to a Seurat object. I don't understand what how this could compromise the identity of study participants? Files could be renamed if they include study ID or participant names. The authors should discuss this with the editor.

Throughout there is a problem with the definition of naïve T cells. Frequently in the text they are described as "naïve, quiescent, and/or stem-like T cells (LEF1, BACH2, IL7R, CXCR4, ZFP36L2)" or variations thereof. However, this is then being abbreviated to "naïve" in other places. The authors need to do further analysis to characterise these cells because there can be substantial differences (or overlap) between naïve/stem T cells with quiescence being a property that either may have and this is crucial to interpreting their findings. See [https://www.cell.com/trends/immunology/fulltext/S1471-4906\(24\)00004-8](https://www.cell.com/trends/immunology/fulltext/S1471-4906(24)00004-8) for example. In particular, I would like to know whether all of these markers are co-expressed within the same cells, or whether there are distinct clusters of "naïve" cells with different expression patterns. Also, do the "naïve" cells show clonal expansions? The expression patterns in the naïve cells are very different at T0 and T1 in fig 2b, suggesting that they are not truly naïve? Oddly, although naïve cells are mentioned in the title they are not in the abstract.

From Fig 1B I do not understand how the participants were age and sex matched, and this is crucial to understanding the differences between controls and children with T1D. Please provide a table listing each donor's characteristics and summarising mean age, sex %. If there are significant differences in the ages of the controls and children with T1D I would consider this a major flaw. In this case, the authors should produce figures showing how their key findings vary by age in controls and children with T1D.

Likewise the manuscript states "This indicates that the selection of healthy controls (e.g., unmatched as in our study or partially matched as in 20) might have a great impact on the results concerning BTN3A2 expression." Does this difference in HLA apply to any of the other DEG you identified?

Minor comments

"On the contrary, the so-called hygiene hypothesis explains the recent outburst of T1D and other autoimmune diseases by the reduction of the infection burden in children 4, 9, 10, 11 ." Please could the authors temper this statement?

"the criterion of partial clinical remission (insulin dose adjusted HbA1c <9)" Please provide a reference for this statement diabetic children Throughout please replace with "children with diabetes". Likewise "T1D patients" to "people with T1D"
"In the next step, we addressed whether Tregs from T1D resemble functionally impaired Tregs from conditions in which their dysfunction is well established and characterized." Please make it clear in the results section where you obtained the samples/data described in this paragraph and how they were integrated with your data.

"We obtained the same results when we compared Tregs from the T1D patients to Treg-like cells in Foxp3-/- mice 56, 57 (Fig. 3H)" I am dubious of the value of including a direct comparison between humans and a mouse model and would prefer that this section is removed.

"The first sample was obtained at a median of 9 (4-24) days after the clinical diagnosis of T1D" This does not match what is said in the results section

Initial scRNAseq experiment: Please put the antibodies/hashtags into a table for clarity

Please state the number of reads/cell for the 10x sequencing

HPAP flow cytometry :please provide a table of exactly which samples were used/excluded

Fig 6- please use asterisks just for significant p values, as including all p values makes this hard to interpret

(Remarks on code availability)

I have read through the notebooks on github and generally find the code to be sufficient and well annotated. There are small areas where something has gone wrong with the text with a mixture of subscripts e.g. 08_Subset_analysis_Treg.ipynb Treg density code chunk. I have not run the code to check it.

As the analysis is in R, it may be more accessible if these were uploaded as Rmd files as well as ipynb, but this is just a suggestion.

The expected rds files are at the zenodo link. Again I have not downloaded and checked them. It would be good if the authors also uploaded rds files that they had processed from other datasets, as these would be a valuable resource for the community.

See also major comment about fastq.

Reviewer #5

(Remarks to the Author)

This manuscript presents the results of an analysis of single-cell gene expression in T cells from healthy donors and children with T1D. In the T1D group, repeated measures were taken close to the time of diagnosis, and then after 1 year. The authors found reduced expression of genes related to effector and cytotoxic T cells in the children with diabetes, which were somewhat normalized by 1 year. Strengths of this manuscript include the relatively large sample size for an scRNA-seq analysis, with repeated measures within a participant. However, there are a number of problems with the analytical methods. My comments on the paper are as follows:

1. The DEG analysis using FindMarkers in Seurat treats each cell as an independent observation and therefore is subject to type I error inflation due to pseudoreplication. In addition, there is no correction for multiple testing, which further inflates the type I error (although the legend for Figure 2A describes a Bonferroni adjustment, but that is not described in the methods, and would also be extremely conservative for an scRNA-seq analysis).
2. The analysis methods do not account for the correlation between T1D and controls induced by performing matching.
3. No power calculation appears to have been performed. This makes it more difficult to interpret findings that were not statistically significant.
4. The study design would be stronger if samples were obtained in the people with T1D prior to diagnosis, so gene expression in the period of time leading up to development of diabetes could be studied.
5. The difference between experiment 1 and experiment 2 is unclear.
6. Genes present in less than 3 cells were removed. It is more common to see this threshold set to 5-10% of cells, in order to remove effects of rare genes.
7. In the methods, it is stated that cells with <200 genes are removed, and in another place it is stated that cells with <500 cells are removed.
8. What criteria were used to select 12 PCs?
9. What is the justification for using the GSE11057 and GSE9650 databases for GSEA rather than a more standard database?
10. Why were fastq files merged prior to HLA typing?
11. No justification is provided for the focus on BTN3A2. Why was this gene scaled differently than the others? The use of a range of -1 to 1 for scaling BTN3A2 seems arbitrary.
12. In the methods for flow cytometry, the authors state that the same number of cells was used in each batch, but they also state that the same number of cells were used and these numbers look quite different.
13. The paper is written in a somewhat non-standard way, with text relevant to the methods included in the results, and very few summary statistics in the text of the results which makes it difficult to evaluate the statements made.
14. Which prior distributions were used in the Bayesian modeling?
15. In the Figure 1D and 1G heatmaps, it could be useful to present the genes in the same order. Dot plots that show both the average level of expression and the percent of cells in which the genes were expressed could be informative. How were these subsets of genes selected?
16. Figure 2a and 2b appear to be a volcano plot that incorporates both the T1D T0 vs Healthy comparison and the T1D T1 vs Healthy comparison, but this figure is not explained very well. Figure 2c could also use additional explanation.
17. The authors claim that this study is the first analysis of repeat measures of single cell transcriptomics in T cells in newly diagnosed patients. At least one other study (PMID: 38967999) seems to meet these criteria?

(Remarks on code availability)

Version 1:

Reviewer comments:

Reviewer #1

(Remarks to the Author)

We are generally satisfied with the authors' responses, except for their response to minor concern #2. The authors identified

a contaminating NK cluster within the Temra cluster (Fig. 5F) in the original manuscript. We suggest the authors remove the NK cluster (Fig. 5F) during quality control rather than in the final analyses because the NK contamination may bias all the results in fig 5. Surprisingly, the authors only removed the NK cluster in Fig 5F while Fig5A-5G remained unchanged. We still believe that all panels in fig 5 should be remade after removing the NK cluster.

(Remarks on code availability)

It's OK

Reviewer #2

(Remarks to the Author)

(Remarks on code availability)

Reviewer #3

(Remarks to the Author)

The authors have worked hard to improve the quality and clarity of the manuscript. In this process they have addressed most of my concerns.

One outlying issue that I feel needs to be mentioned is the HLA-matching. While I agree that 'perfect' HLA matching is not possible, it is possible to match for the HLA alleles of known significance for T1D (HLA-DR4, DQ8, DR3 and DQ2). I also accept that this 'matching' is not perfect it is nonetheless an attempt to avoid being misled by difference attributable to HLA alleles. Not doing this leaves open the possibility that the differences observed between individuals with and without T1D may simply be attributable to the HLA alleles within each group, not their T1D status.

(Remarks on code availability)

Reviewer #4

(Remarks to the Author)

Thank you for your revisions which have answered all of my queries except for the question below on raw data availability. Although I would still expect the raw data to be made available I am also happy to defer to the editor on what is the minimum acceptable to the journal.

“Processed data required to generate figures for the manuscript are available on Zenodo: DOI: 10.5281/zenodo.14222418 Raw data were not deposited to protect the identity of study participants.” Under FAIR data principles, I would expect to have access to the raw data upon point of publication e.g. fastq files as much information is lost by running the cellranger pipeline and conversion to a Seurat object. I don't understand what how this could compromise the identity of study participants? Files could be renamed if they include study ID or participant names. The authors should discuss this with the editor. We thank the reviewer for this comment. The privacy challenges associated with sharing next-generation sequencing data have been discussed recently in multiple studies and reviews (e.g., PMID: 34759381, PMID: 39362221). Raw sequencing data can be effectively used to identify individual donors based on SNP (PMID: 38622708). Due to European privacy laws (GDPR), individual-level data from living human subjects cannot be made fully public, similar to other studies published in Nature Communications (e.g., PMID: 38714671). Our raw data may be made available to qualified researchers for the sole purpose of

13 replicating procedures and results, provided that data transfer complies with GDPR and is approved by the relevant ethical committees. We defer to the editor's guidance on data availability requirements.

(Remarks on code availability)

As previously, I have checked the code and it looks good but I did not run it myself

Reviewer #5

(Remarks to the Author)

Thank you for the thoughtful responses to my comments. My remaining suggestion would be to include something in the methods about the use of the Bonferroni correction and the fact that study participants were frequency matched rather than 1:1 matched.

(Remarks on code availability)

REVIEWER COMMENTS

Reviewer #1 (Remarks to the Author):

It is important to profile T cells in type 1 diabetes (T1D) since T1D is thought as an autoimmune disease. The study by Veronika et al. investigated the T cell subsets and their features in T1D by comparing with healthy donors, revealing a disrupted balance in T cell phenotypes across several T cell subsets in T1D patients. While the topic is intriguing, the study's persuasiveness is somewhat diminished by the presentation of the results. The authors should conduct more robust experiments or perform a more rigorous analysis to substantiate their findings.

We thank the reviewer for recognizing the importance of our study and for the constructive comments. As detailed below, we addressed these points by clarifying our analyses, revising figures and text, and performing additional supporting analyses to improve the rigor and clarity of the manuscript.

Major concerns:

1. The authors performed batch correction on six batches from initial scRNAseq dataset and final scRNAeq dataset. However, there seems to be a significant batch effect between the initial and final datasets after batch effect correction (Figure S2J), and the batch effect still could significantly bias the results. It is much better to project all the cells on the UMAP plot to examine the integrated results, which could help we understand whether the T cell subsets are common or sample specific.

We thank the reviewer for this important point. We are sorry that we were not able to explain what the data in Fig. S2J represent in a clearer way, which led to this misunderstanding. The initial scRNAseq experiment (batches 1-2) contained all CD4⁺ or CD8⁺ T cells, whereas in the final experiment (batches 3-6) non-naïve T cells were enriched in the 1:5 naïve:non-naïve ratio. For this reason, it is expected that the integration does not normalize the differences between initial and final experiment. The fact that all the batches of the final experiment are well corrected, but still distinct from the initial datasets actually shows that the integration method worked well and did not induce over-integration. We did our best to explain this in the revised version of the manuscript: we added a new supplementary Fig. S3 showing that the integration effectively removed differences between batches, but not between enriched and non-enriched samples. Moreover, we added a new Supplementary Table S3 specifying the counts of samples in individual batches and we revised the text of the Results and Methods sections accordingly.

We added UMAPs with labeled cells based on the batches showing the enrichment of non-naïve T cells in the final experiment according to the expectation (new Fig. S3I).

2. The authors examined the fraction of T cell subsets in healthy donor and T1D patients and their change mainly using FACS sorting (Fig 2D, 2F, 3E-F, 4F-H, 5H, 6F-G). However, the authors did not examine the fraction of T cell subsets in scRNAseq data. Whether the FACS-results and scRNAseq results are consistent?

We are thankful for these comments. As we decided to enrich non-naïve cells in the final experiment, the fractions of the T-cell subsets are altered significantly (see newly added Fig. S3J) and therefore the interpretation of their abundance is difficult. Nevertheless, we quantified cluster frequencies and evaluated the changes by frequentist and Bayesian statistics (original Fig. S5 which is now Fig S6). To show that in the initial scRNAseq experiment (non-enriched samples), the FACS and scRNAseq data are consistent, we added a new Fig. S10A which shows the correlations of the frequencies of naïve CD4⁺ T

cells, CD4⁺ Treg cells, naïve CD8⁺ T cells and CD8⁺ $\gamma\delta$ Tgd cells measured by FACS and scRNAseq (new Fig. S10A).

3. The authors identified DEGs and signals of CD8⁺T cells (or CD4⁺T cells) between T1D patients and healthy donors without considering T cell subsets (Fig 2A). The author further identified the DEGs of each T cell subset between T1D patients and healthy donors (Figure S6B). However, it is still unclear whether the identified DEGs of these T-cell subsets are consistent with each other or T cell subset specific? The authors could make a more explicit figure to clearly show DEGs and signals are shared by multiple T cell subsets or specific T cell subsets.

We thank the reviewer for this important point. We agree that the original Figure S6B was difficult to interpret, largely because we tried to summarize DEGs across all 14 subpopulations.

To address this, we revised and expanded Figure S6 by splitting it into two new supplementary figures: Fig. S8 focusing on the contrasts between T1D patients and healthy donors and Fig. S9 focusing on the DEG between T1D patients with/without ketoacidosis and T1D patients with/without partial remission. This allowed us to allocate more space to analysis of DEG in subpopulations of CD8⁺ T cells (Fig. S8A-B) and CD4⁺ T cells (Fig. S8C-D).

In the new Fig. S8: Panels A and C show heatmaps highlighting the top DEGs for each subpopulation of CD8⁺ T cells (Fig. S8A) or CD4⁺ T cells (Fig. S8C). We focused on the top 5 DEGs per subset to ensure that both shared and subset-specific signals are clearly visible. For example, TNF is upregulated in healthy donors versus T1D T0 in CD8⁺ Temra cells; IFI27 and CD38 are upregulated in T1D T1 versus T0 in proliferating cells (Fig. S8A); and GZMK is upregulated in T1D T0 versus healthy in CD4⁺ Temra cells (Fig. S8C).

Panels B and D show heatmaps quantifying the overlap of the top 100 up- and down-regulated genes across subpopulations, illustrating which signatures are shared and which are subset-specific. This analysis shows that naïve and memory subsets largely share similar gene expression patterns, while some effector or unconventional subsets show unique patterns.

We believe these new figures address the reviewer's request by explicitly visualizing shared and subset-specific DEGs.

4. The feature, repeatability and functions of these Treg subsets in T1D are not well demonstrated. E.g. the authors claim that Treg1 has phenotypes of naïve Treg. However, Treg1 also highly expressed IL4R, IL7R, LAG3, GZMK and CD226 (Fig 3B) which are memory T or effector T specific genes. How about the fraction of Treg1 in each sample based on scRNAseq data? Whether Treg1 are present in each sample? Is Treg1 more likely to differentiate into Treg4 in healthy donors than in T1D using trajectory analysis?

We thank the reviewer for this insightful comment. We agree that in conventional T cell subsets, markers such as IL4R, CD226, GZMK, and LAG3 are not typically associated with a naïve phenotype. However, we would like to highlight that Treg1 cells also express classic naïve markers, including TCF7, LEF1, SELL, and CCR7. IL7R is indeed expressed on naïve T cells and upregulated on central memory cells, but downregulated in effectors, which supports our characterization. We have clarified in the revised manuscript that Treg1 cells co-express genes associated with a naïve phenotype as well as genes linked to Treg dysfunction (e.g., IL4R, CD226, GZMK), while lacking key Treg functional markers such as FOXP3, CTLA4, IL2RA, and TIGIT.

Regarding their distribution, Treg1 cells are present in each sample, as shown in the frequency plots in Fig. S12A.

To further investigate the relationship between Treg subsets, we performed a pseudotime analysis using the slingshot package (PMID: 29914354) and validated that the inferred trajectory reflects Treg differentiation (new Fig. 3G-L). We additionally used the condiments package (PMID: 38280860) to test for differences in progression along the trajectory between groups. This analysis revealed that Tregs are more activated in healthy donors compared to individuals with T1D, suggesting differential activation or differentiation states.

We have updated the manuscript text accordingly to reflect these findings.

5. It seems that TR3-56 cells are also located on conventional CD8+ T cells (Fig4B). What are the differences of the TR3-56 cells on conventional CD8+ T cells and MAIT/ $\gamma\delta$ T?

We thank the reviewer for this question. To clarify this point, we extracted TR3-56 cells from each parental cluster (conventional CD8⁺ T cells, MAIT cells, $\gamma\delta$ T cells, and NK cells) and compared their TR3-56 scores (new Fig. 4D). This analysis showed that TR3-56 cells originating from MAIT and $\gamma\delta$ T cell clusters had the highest TR3-56 scores, indicating that these might be the bona fide TR3-56 cells. We also performed DEG analysis to study the differences between TR3-56 cells coming from the different parental cluster, which revealed the signature genes of the parental clusters, such as CD8A and CD8B in conventional CD8⁺ T cells, IL7R, KLRB1 in MAIT cells and TRDC, TRGV2 in Tgd cells (see Response Figure 1 below).

Response Figure 1. Heatmap of differentially expressed genes between TR3-56 cells coming from clusters of Conventional CD8 T cells, gd T cells and MAIT cells.

Minor concerns:

1. The expression of effector genes (NKG7, GNLY, CCL4, CCL5, CX3CR1, TNFRSF4, TNF, and TNFRSF9) in EGAD00001005768 are different the other data. The authors should examine and discuss this inconsistency.

We thank the reviewer for this comment. We agree that the expression of certain effector genes in the EGAD00001005768 dataset, as well as in the syn53641849 dataset, differs from our data and other public datasets. A direct comparison of results from different studies is crucial for general understanding of the problem. Yet, this is not routinely done because of several challenges such as technical differences and biases among particular studies. For this reason, we compared our results with all relevant transcriptomic studies with a similar design. This analysis confirmed that our results are consistent with the majority of these studies. An unanimous consensus across studies was not anticipated because of the above-mentioned discrepancies. We believe that a detailed exploration of the reasons for this discrepancy is outside the scope of our current study.

We looked at the genes which showed differences in the studies mentioned by the reviewer in a closer detail by performing a PCA analysis of normalized log fold changes of all genes across studies (see Response Figure 2 below). This analysis shows that these genes cluster together (A), indicating that the differences are not specific to individual genes but rather reflect broader technical variability in sample processing across studies (B). We defer to the editor's decision regarding whether further discussion of this point is necessary.

Response Figure 2. PCA of gene expression fold changes across datasets. (A) PCA showing the distribution of genes based on their normalized log-fold changes across studies. (B) PCA biplot showing the contribution of individual datasets to gene-level variability, with arrows indicating the direction and strength of dataset-specific effects.

2. Contaminating NK cluster (Fig. 5F) should be removed at the quality control or cell annotation rather than the final analyses.

We removed the NK cluster prior to the final analyses.

3. Some errors should be checked and corrected, such as

1) The term "NK T cells" should be clarified. Is it referring to NK cells or Natural Killer T (NKT) cells? (Figures S3A-D)

We thank the Reviewer for spotting this typo. It is NK cells. This is corrected in the main text.

2) The expression of CD79A, a B cell marker, by iNKT cells is puzzling and requires an explanation.

We thank the reviewer for raising this interesting point. Although expression of CD79A by iNKT cells is indeed puzzling, it has been reported previously in a subset of iNKT cells from human peripheral and cord blood (PMID 40305288 and PMID 32528956). As a further validation, we found increased CD79A expression in iNKT cells in RNAseq data from the Immgen database (see figure below). Thus, we believe that expression of CD79A in some iNKT cells is real. The biological relevance of this is not known and is questionable for two reasons: 1) Jayasinghe et al. (PMID 40305288) showed that CD79A protein can not be detected on the surface of iNKT cells, 2) compared to B cells, the expression is ~ 100 times lower than in B cells (see Response Figure 3 below). Nonetheless, we believe that CD79A is still useful as a specific marker that distinguishes iNKT cells from conventional T cells, thus we decided to keep it in the heatmap in the Fig. S5C.

Response Figure 3. Expression of Cd79a by subsets of T cells, B cells and NKT cells. Data from the ImmGen database (<https://www.immgen.org/>) were visualized using the Gene Skyline application with reference data from ImmGgen ULI RNASeq.

3) Legends should be added to figures for clarity (e.g., Figs 6B-6C, S9A, F.).

We thank the reviewer for noticing this. We added the labels to the panels mentioned.

Reviewer #1 (Remarks on code availability):

It is OK

Reviewer #2 (Remarks to the Author):

We are thankful for the contribution of this Reviewer to the Peer-Review of our manuscript.

Reviewer #3 (Remarks to the Author):

The manuscript by Niederlova et al describes the analysis of T-cell population from children with and without T1D at two time points a year apart. The manuscript points to some novel aspects of human T cell biology relating to T1D. Several parts of the manuscript could be improved. These are listed below.

We are thankful for the generally positive evaluation of our work and for insightful suggestions to improve the manuscript.

Excerpts from the text are in italics, my comments are in regular text

Major points:

1. The title not clear. I assume it is the cells, not the phenotypes, that are in imbalance.

We agree and changed the title accordingly. The new title is 'Imbalance of stem-like and effector T cell states in early type 1 diabetes across conventional and regulatory subsets'.

2. To compare the T-cell compartments in healthy donors and T1D children... It is more respectful of people with T1D to refer to people, or children in this case, with and without T1D.

We agree with the reviewer and thank them for pointing it out. We revised the manuscript accordingly.

3. The composition and the matching of the individuals in the cohorts is critical to this study. 30 T1D vs 13 HC, but not HLA matched? What about age and sex?. This is shown diagrammatically in Figure1, but needs to be stated explicitly and clearly in the Results section.

We thank the reviewer for this comment. We revised the manuscript by adding the explicit characterization of our cohort in the Results section and added plots (Fig. S1A-B) and tables with summary statistics (Tables S1 and S2). To evaluate the age and sex differences between T1D and healthy donors, we performed standardized mean difference (SMD) analysis (Fig. S1B), which revealed that our cohort is well-balanced in terms of the age, but not the sex of participants. We clearly stated this in the Results section and we validated the main results separately for each age group and for each sex group (Fig. S7G-H).

Regarding HLA matching, we did not use HLA-matched controls for two main reasons. First, it is extremely difficult to recruit fully HLA-matched healthy donors, and partial matching can introduce additional confounders. Second, our study aimed to capture global disease-associated changes rather than isolate HLA-specific effects. Importantly, a key strength of our design is the inclusion of repeated samples from the same T1D individuals over time, where HLA background is constant and does not influence intra-individual comparisons.

4. First, we performed an initial scRNAseq analysis of CD4+ and CD8+ lymphocytes from 16 samples... Were these children with, or without T1D? How were these 16 samples chosen? Did the CD45RO/RA+ correlate with the subject's age?

We thank the reviewer for this comment and apologize for the lack of clarity. The initial set of 16 samples included both children with T1D (n = 12) and without T1D (n = 4). These samples represented the first

available samples collected at the start of the study (thus not pre-selected or randomized). We have now specified this in the Results section and added Table S3 summarizing sample numbers, donor counts, and enrichment strategy for both the initial and final experiments.

Regarding the correlation between CD45RA/RO expression and age, we performed this analysis and observed an expected inverse correlation between CD45RA expression and age in CD8⁺ T cells (see figure below). We are happy to share this result directly with the reviewer (see Response Figure 4 below), but have not included it in the main manuscript as it does not directly impact the primary conclusions of our study.

Correlation of average PTPRC-RA expression per sample in CD8⁺ T cells with age

Response Figure 4. Correlation of the expression of CD45 (PTPRC) isoform RA with age. Expression in each sample from pseudobulked CD8⁺ dataset by age (left) and grouped by age groups (right).

5. Can the authors validate that impact of batch specific effects did not impact their analysis and interpretation of the data? Supplementary materials show plots, but can this reduction in batch effects be quantified more objectively?

To address this and related reviewers' comments, we substantially revised the manuscript. We now better explain the key difference between the initial and final experiments. In the initial experiment, we analyzed all CD4⁺ or CD8⁺ T cells, while in the final experiment, we enriched the non-naïve population by cell sorting in a 1:5 naïve:non-naïve ratio. This difference, rather than technical batch effects, explains the apparent separation in the original PCA plots (original Fig. S2J). We clarified the experimental design and added new plots (new Supplementary Fig. S3) illustrating the distinct sorting protocols.

To objectively quantify batch effect reduction, we compared PCA distances between samples from (1) the same batch, (2) different batches with the same enrichment, and (3) different batches with different enrichment, both before and after integration (Fig. S3C-D, G-H). The integration clearly reduced batch-related distances while preserving biologically relevant differences.

Finally, we performed the *k*-nearest-neighbor Batch-Effect Test (kBET) (PMID: 30573817) specifically tailored to estimate differences between batches of samples in scRNAseq data, which further supports the effectiveness of our integration (see Response Figure 5 below).

Response Figure 5. K-nearest-neighbor Batch-Effect Test related to figure S3.

6. Have the authors stratified for the age of the participants? Could the differences described be attributable to the age of the participants more than their diagnosis with, or without, T1D?

We thank the reviewer for this important point. The groups of children with and without T1D were balanced in terms of age distribution, as confirmed by our newly added standardized mean difference (SMD) analysis (Fig. S1B). We agree that age is particularly relevant when interpreting naïve versus effector T-cell phenotypes, which are known to vary with age. To address this, we complemented the original GSEA with an additional approach using decoupleR, which explicitly incorporates the sign and weight of network interactions, providing a more robust, donor-level assessment of transcriptional signatures (Fig. 2D). We visualized the results of these analyses separately within each age group (New Fig. S7G-H). These analyses revealed that the observed differences between T1D patients and healthy donors were consistent across age groups, supporting that our main findings are not driven by age differences.

7. The first paragraph of the discussion is irrelevant and should be deleted. If the authors feel this information should be included it should be in the Introduction.

We agree with the reviewer. We deleted the first paragraph of the discussion.

Minor points

1. In my opinion it is not true to say that there has been a ‘steep increase in T1D incidence...’ (Introduction paragraph 1). This should be reworded to something less hyperbolic.

We agree with the reviewer. We revised the sentence by removing the word ‘steep’.

2. (Introduction paragraph 2) Despite numerous studies in the field...please cite some of these studies

We added citations to a few selected studies.

3. The Network for Pancreatic Organ Donors with Diabetes. Please include the acronym nPOD.

We thank the reviewer for this suggestion. As the full name of the organization appears only once in the manuscript, we did not initially include the acronym. We recognize that the Network for Pancreatic Organ Donors with Diabetes is widely known under its acronym nPOD. Therefore, we have added the acronym in the revised version of the manuscript. However, we defer to the editorial decision regarding its inclusion to fully align with the journal's style guidelines.

4. We used a balanced medium-sized cohort...it is not clear what 'balanced' refers to here. Medium-sized is also subjective, state the number of individuals in each cohort.

In the revised manuscript, we now explicitly state the exact cohort sizes rather than using the subjective terms "balanced" and "medium-sized."

5. Blood of 30 donors with T1D was taken 4-9 (median 9) days...this is not possible. If the range is 4-9, the median cannot be 9.

Thank you for catching this inconsistency. The statement in the Results section was a typo, and we have corrected it to match the correct information provided in the Methods section.

6. The T1D patients showed higher overall blood lymphocytes and lower neutrophils at T0 in comparison to T1 and healthy donors (Fig. S1C). Give the data and the statistics in the text.

We agree with the reviewer. We added the data and the statistics to the revised version of the Results section.

7. We compared the proportions of particular subsets using both frequentists...please correct typo.

We revised the manuscript accordingly.

8. ...which could be further divided to naïve, central memory (Tcm), Treg, Th1/Th17, Th2, Temra, ISAGhi. Please define ISAGhi and Tsig in the text,.

We revised the manuscript accordingly.

9. How many participants were included in the HPAP study which the authors reanalyzed?

We included samples from 53 HPAP donors across three staining panels: 41 samples in Panel 1 – CD4 phenotyping panel, 19 samples in Panel 2 – CD8 phenotyping panel and 26 samples in Panel 3 – CD8 phenotyping panel focused primarily on antigen-specific cells. We revised the manuscript by adding the numbers of participants to the Results section. In addition, we added a new Table S6 indicating the specific HPAP samples used in the study.

10. First, we observed that our healthy donors generally followed the common distribution of HLA alleles in the Czech population 45 with the exception of overrepresented HLA-A*02:01, whereas most T1D children carried the respective susceptible alleles. This sentence is very unclear. The non-T1D subjects

were more frequently A2+ than expected?...Even though there were only 13? What is meant by 'respective susceptibility alleles- HLA-DR4-DQ8, DR3-DQ2?

*We thank the reviewer for this helpful comment. We agree that the mention of HLA-A*02:01 could be confusing (it was just an observation, it was not statistically significant) and anyways, it was not essential for our analysis. Therefore, we have removed this detail from the manuscript to improve clarity and focus. We now describe only the general HLA distribution in healthy donors and explicitly list the T1D susceptibility alleles (HLA-DR3-DQ2 and HLA-DR4-DQ8) carried by most T1D children.*

Reviewer #4 (Remarks to the Author):

- What are the noteworthy results?

Niederlova et al have produced an interesting piece of research examining differences in T cell populations between children without T1D, children newly diagnosed, and those one year after diagnosis. They describe in great detail changes in effector T cell populations. A strength of the work is confirmation of the findings by re-examination of previously published scRNAseq and flow cytometry data sets.

- Will the work be of significance to the field and related fields? How does it compare to the established literature? If the work is not original, please provide relevant references.

This work is useful to the field as we attempt to understand phenotypes of both autoantigen- specific and non specific T cells in T1D. Other papers have performed scRNAseq and flow cytometric analysis of T cells before (and these are well described in the introduction of this manuscript), but the focus on children at point of diagnosis and after one year adds value in this study.

- Does the work support the conclusions and claims, or is additional evidence needed?

In general yes, but see major comments below on additional analysis required

- Are there any flaws in the data analysis, interpretation and conclusions? - Do these prohibit publication or require revision?

Yes please see the comments below that require revision

- Is the methodology sound? Does the work meet the expected standards in your field?

Yes

- Is there enough detail provided in the methods for the work to be reproduced?

For the wet lab work- yes.

For the data analysis -yes, but see comments on code

We are thankful for the overall positive evaluation of our study and for insightful comments that helped us to improve the manuscript during revisions.

Major comments

“Processed data required to generate figures for the manuscript are available on Zenodo: DOI: 10.5281/zenodo.14222418 Raw data were not deposited to protect the identity of study participants.” Under FAIR data principles, I would expect to have access to the raw data upon point of publication e.g. fastq files as much information is lost by running the cellranger pipeline and conversion to a Seurat object. I don’t understand what how this could compromise the identity of study participants? Files could be renamed if they include study ID or participant names. The authors should discuss this with the editor.

We thank the reviewer for this comment. The privacy challenges associated with sharing next-generation sequencing data have been discussed recently in multiple studies and reviews (e.g., PMID: 34759381, PMID: 39362221). Raw sequencing data can be effectively used to identify individual donors based on SNP (PMID: 38622708). Due to European privacy laws (GDPR), individual-level data from living human subjects cannot be made fully public, similar to other studies published in Nature Communications (e.g., PMID: 38714671). Our raw data may be made available to qualified researchers for the sole purpose of

replicating procedures and results, provided that data transfer complies with GDPR and is approved by the relevant ethical committees. We defer to the editor's guidance on data availability requirements.

Throughout there is a problem with the definition of naïve T cells. Frequently in the text they are described as “naïve, quiescent, and/or stem-like T cells (LEF1, BACH2, IL7R, CXCR4, ZFP36L2)” or variations thereof. However, this is then being abbreviated to “naïve” in other places.

We thank the reviewer for highlighting this important point regarding our use of the term "naïve". We acknowledge that there may have been a partial misunderstanding caused by our use of the term "naïve" in two contexts: 1. to describe a specific cell cluster and also 2. to refer to a broader transcriptional program characterized by quiescence and stem-like features. To clarify, we have now:

- 1. Referred to the cluster identified by canonical naïve markers (e.g., CCR7, LEF1, SELL) as “Naïve T cells” (Fig. 1).*
- 2. Referred to the differentially expressed gene program (LEF1, BACH2, IL7R, etc.) as a "stemness-associated gene signature” (Fig. 2), to better reflect its broader expression beyond strictly naïve cells.*

The authors need to do further analysis to characterise these cells because there can be substantial differences (or overlap) between naïve/stem T cells with quiescence being a property that either may have and this is crucial to interpreting their findings. See

[https://www.cell.com/trends/immunology/fulltext/S1471-4906\(24\)00004-8](https://www.cell.com/trends/immunology/fulltext/S1471-4906(24)00004-8) for example.

We thank the reviewer for their suggestions. We performed additional analyses using a bulk RNA-seq reference dataset of sorted naïve, stem cell memory, central memory, effector memory and Temra T cells (PMID: 35263570). By annotating our datasets with these reference signatures, we confirmed that the "Naïve T cell cluster" corresponds to true naïve cells, while stem cell memory cells are enriched in clusters annotated as memory cells (new Fig. S4G-H, S5H-I).

To further investigate the relationship between naïve and stemness-associated signatures, we complemented the original GSEA with an additional approach using decoupleR and the aforementioned reference dataset. DecoupleR provides a more robust, donor-level assessment of transcriptional signatures and revealed enrichment of both naïve and stem cell memory signatures in T1D samples, along with a decrease in Temra signature (Fig. 2D). We have rephrased the text to reflect the finding that the enriched genes are shared between naïve and stem cell memory subsets, and we now consistently refer to this as the "stemness-associated gene signature."

In particular, I would like to know whether all of these markers are co-expressed within the same cells, or whether there are distinct clusters of “naïve” cells with different expression patterns.

We examined the expression of SELL, LEF1, BACH2, IL7R, CXCR4, and ZFP36L2 in different subclusters of naïve CD4+ and CD8+ T cells. Overall, the naïve clusters are homogeneous, with the exception of a few $\gamma\delta$ T cells among naïve CD4+ T cells and a subset of cells expressing interferon-stimulated genes among naïve CD8+ T cells. The key markers appear to be co-expressed across all naïve subclusters (see Response Figure 6 below).

Response Figure 6. Expression of naïve markers in subclusters of naïve T cells. Upper panels – CD4+ T cells, lower panels – CD8+ T cells.

The expression patterns in the naïve cells are very different at T0 and T1 in fig 2b, suggesting that they are not truly naïve?

The reviewer is referring to Fig. 2B, which shows a high expression of the stemness-associated gene signature in children with T1D at T0 compared to T1. We would like to clarify, that the heatmap shows the expression of a stemness-associated gene signature across all cells, not specifically within the naïve T cell cluster. We interpret this as a partial normalization or recovery of T cell phenotype following treatment initiation, as the T1 profile is closer to that of healthy donors.

Also, do the “naïve” cells show clonal expansions?

Clonality analysis confirmed that there is no clonal expansion in the clusters of Naïve T cells (Fig. 1E and 1H).

Oddly, although naïve cells are mentioned in the title they are not in the abstract.

We agree with the reviewer and thank them for noticing this. We revised the abstract accordingly.

From Fig 1B I do not understand how the participants were age and sex matched, and this is crucial to understanding the differences between controls and children with T1D. Please provide a table listing each donor's characteristics and summarising mean age, sex %. If there are significant differences in the ages of the controls and children with T1D I would consider this a major flaw. In this case, the authors should produce figures showing how their key findings vary by age in controls and children with T1D.

We thank the reviewer for this important comment. We fully agree that accounting for age (and sex) is crucial when comparing T1D and control cohorts. In the revised manuscript, we now include summary tables (Tables S1 and S2) detailing participant characteristics, including age, sex, BMI, fasting C-peptide, HbA1c, and HLA genotypes for T1D donors at both T0 and T1, as well as healthy controls. Individual-level metadata are available in the R objects deposited on Zenodo.

We show that the T1D and healthy donor cohorts were comparable in age distribution, as confirmed by descriptive statistics (new Table S1, new Fig. S1A) and standardized mean difference (SMD) analysis (new Fig. S1B). However, the sex distribution differed, with a higher proportion of males in the T1D cohort (67%) compared to the healthy donors (33%) (Fig S1B, Table S2). To address this, we validated key findings separately in males and females, demonstrating that sex is unlikely to confound our results (Fig. S7G-H).

Likewise the manuscript states “This indicates that the selection of healthy controls (e.g., unmatched as in 20) might have a great impact on the results concerning *BTN3A2* expression.” Does this difference in HLA apply to any of the other DEG you identified?

*We thank the reviewer for this important point. We also examined *TNF*, the only other differentially expressed gene in our dataset located within the (extended) MHC region aside from *BTN3A2*, *BTN3A3* and *HLA* genes. In contrast to *BTN3A2*, we did not observe a significant association of *TNF* expression with HLA haplotype in our dataset (new Response Figure 7 below). This suggests that *BTN3A2* is uniquely impacted by HLA background among our DEGs, while other DEGs, including *TNF*, are unlikely to be confounded by HLA differences.*

Response Figure 7. Expression of *BTN3A2* and *TNF* in participants grouped by HLA haplotype. Each dot represent a pseudobulked sample of CD8+ T cells or CD4+ T cells from one participant.

Minor comments

“On the contrary, the so-called hygiene hypothesis explains the recent outburst of T1D and other autoimmune diseases by the reduction of the infection burden in children 4, 9, 10, 11 .” Please could the authors temper this statement?

We revised the manuscript accordingly.

“the criterion of partial clinical remission (insulin dose adjusted HbA1c <9)” Please provide a reference for this statement

We added citations to the original definition (PMID: 19435955) and for the results from a validation cohort (PMID: 25287319).

diabetic children Throughout please replace with “children with diabetes”. Likewise “T1D patients” to “people with T1D”

We thank the reviewer for pointing it out. We revised the manuscript by using the term “children with T1D”.

“In the next step, we addressed whether Tregs from T1D resemble functionally impaired Tregs from conditions in which their dysfunction is well established and characterized.” Please make it clear in the results section where you obtained the samples/data described in this paragraph and how they were integrated with your data.

We added this information to the Results section and made the reanalyzed Treg datasets available on Zenodo.

“We obtained the same results when we compared Tregs from the T1D patients to Treg-like cells in Foxp3^{-/-} mice 56, 57 (Fig. 3H)” I am dubious of the value of including a direct comparison between humans and a mouse model and would prefer that this section is removed.

We thank the reviewer for this comment. We would like to clarify that we did not directly compare human and mouse data. Instead, we analyzed how human orthologues of genes differentially expressed in Treg-like cells from Foxp3^{-/-} mice versus wild-type mice are expressed in Tregs from children with and without T1D. We believe this analysis is justified, particularly since we observed similar patterns in humans with FOXP3 loss-of-function mutations. However, we agree that this comparison may not be essential for all readers. Therefore, we have revised the figure and moved the mouse comparison to the supplemental figure (Fig S12C-D).

“The first sample was obtained at a median of 9 (4-24) days after the clinical diagnosis of T1D” This does not match what is said in the results section

Thank you for catching this inconsistency. The statement in the Results section was a typo, and we have corrected it to match the correct information provided in the Methods section.

Initial scRNAseq experiment: Please put the antibodies/hashtags into a table for clarity

We thank the reviewer for this suggestion. The antibodies and hashtags used in the initial scRNA-seq experiment are described in detail in the Methods section. We agree that including this information in a table may improve clarity for some readers, and we defer to the editor's recommendation on whether an additional table should be included. We are happy to provide it upon request.

Please state the number of reads/cell for the 10x sequencing

In all sequencing batches, we targeted at least 35,000 reads per cell for gene expression libraries, as recommended by the 10x Genomics protocols, along with an additional 3,500 reads per cell for the V(D)J libraries and 2,000 reads per cell for the cell surface protein libraries. In practice, our sequencing yielded an average of 45,745 reads per cell for gene expression libraries and 3,712 reads per cell for V(D)J libraries (Table S4). We have now included this information in the Methods section and provide additional detailed metrics in the new Table S4.

HPAP flow cytometry :please provide a table of exactly which samples were used/excluded

We have provided this table as a new supplementary Table S6.

Fig 6- please use asterisks just for significant p values, as including all p values makes this hard to interpret

Thank you for this suggestion. The guidelines of Nature Communications indicate: "Where relevant, provide exact values for both significant and non-significant P values." For this reason, we have included all p-values in the figure. To improve readability while complying with these guidelines, we have highlighted significant p-values in red in the revised Fig. 6.

Reviewer #4 (Remarks on code availability):

I have read through the notebooks on github and generally find the code to be sufficient and well annotated. There are small areas where something has gone wrong with the text with a mixture of subscripts e.g. 08_Subset_analysis_Treg.ipynb Treg density code chunk. I have not run the code to check it.

We are thankful for the positive evaluation. We have revised the mentioned issues.

As the analysis is in R, it may be more accessible if these were uploaded as Rmd files as well as ipynb, but this is just a suggestion.

We agree with the reviewer that R analyses are more often reported as Rmd files. However, as we performed all the analyses in the Jupyter Notebook, we do not have these available.

The expected rds files are at the zenodo link. Again I have not downloaded and checked them. It would be good if the authors also uploaded rds files that they had processed from other datasets, as these would be a valuable resource for the community.

We agree with the reviewer. We uploaded all datasets from our study as well as from other studies to Zenodo.

See also major comment about fastq.

Reviewer #5 (Remarks to the Author):

This manuscript presents the results of an analysis of single-cell gene expression in T cells from healthy donors and children with T1D. In the T1D group, repeated measures were taken close to the time of diagnosis, and then after 1 year. The authors found reduced expression of genes related to effector and cytotoxic T cells in the children with diabetes, which were somewhat normalized by 1 year. Strengths of this manuscript include the relatively large sample size for an scRNA-seq analysis, with repeated measures within a participant. However, there are a number of problems with the analytical methods. My comments on the paper are as follows:

We are thankful for the overall positive evaluation of our manuscript and for the insightful comments that helped us to improve the manuscript.

1. The DEG analysis using FindMarkers in Seurat treats each cell as an independent observation and therefore is subject to type I error inflation due to pseudoreplication. In addition, there is no correction for multiple testing, which further inflates the type I error (although the legend for Figure 2A describes a Bonferroni adjustment, but that is not described in the methods, and would also be extremely conservative for an scRNA-seq analysis).

We thank the reviewer for raising this important point regarding type I error inflation due to pseudoreplication in cell-level differential expression analysis using Seurat's FindMarkers function. In response, we performed a new sample-level analysis using pseudobulk aggregation, in which counts were aggregated per sample, and differentially expressed genes and log fold changes were calculated using DESeq2. This analysis showed a strong correlation with the original cell-based analysis performed with Seurat (Fig. S7A-B).

Interestingly, the DESeq2-based pseudobulk analysis also revealed genes associated with NFAT–calcium signaling that we have not previously appreciated, further strengthening our conclusions regarding effector phenotypes and providing additional mechanistic insights. We would like to thank the reviewer for this helpful suggestion.

In the revised manuscript, we now show that key genes representing effector-, stemness-, and NFAT-associated signatures were identified consistently by both cell-level and pseudobulk analyses (Fig. S7C, E), and demonstrated statistically significant differences at the sample level (Fig. S7D, F).

Regarding multiple hypothesis correction, as noted in the figure legend, we used Bonferroni correction on all features in the dataset, which is the default correction method implemented in Seurat. We acknowledge that this approach is highly conservative; however, we chose it intentionally to mitigate type I error inflation due to pseudoreplication, as the reviewer rightly pointed out.

To further strengthen our findings at the sample level, we complemented the original GSEA analysis, which relied on cell-level log fold changes calculated by Seurat's FindMarkers function. Specifically, we introduced an additional approach that quantifies the similarity of each donor's gene expression profile to reference contrasts between naïve and memory cells from a previously published dataset (GSE179613), using decoupleR. Unlike GSEA, this method explicitly incorporates the sign and weight of network interactions, providing a more robust and donor-level assessment of transcriptional signatures.

2. The analysis methods do not account for the correlation between T1D and controls induced by performing matching.

We thank the reviewer for this insightful comment. In the original version of the manuscript, we used the term “age-matched healthy donors”, which may have been misleading. We did not perform individual (e.g., 1:1) matching between T1D and control participants. Rather, we selected a control group that was balanced in age distribution compared to the T1D cohort. This was confirmed using standardized mean difference (SMD) analysis (new Fig. S1B), which demonstrated minimal age imbalance between groups. Since no individual matching was applied, there is no induced correlation structure that needs to be accounted for in the statistical analysis. Furthermore, to ensure that our main results are not confounded by age or sex, we validated key findings separately by age group and sex (new Figs. S7G-H), confirming that these variables do not drive our conclusions. We have revised the manuscript to clarify these points in both the main text and Methods section.

3. No power calculation appears to have been performed. This makes it more difficult to interpret findings that were not statistically significant.

We agree with the reviewer’s comment. When we began collecting samples, there were no published scRNA-seq datasets on T1D and very few robust bulk transcriptomic studies available. Consequently, it was not possible to reliably estimate variability or effect size a priori, and therefore we did not perform a formal power calculation at the outset. We explicitly discuss this limitation in the Nature Portfolio Reporting Summary, and we would be happy to include it in the manuscript text as well if requested by the editor.

4. The study design would be stronger if samples were obtained in the people with T1D prior to diagnosis, so gene expression in the period of time leading up to development of diabetes could be studied.

We agree with the reviewer that studying gene expression during the preclinical phase of T1D would strengthen the design and provide valuable insight into early disease mechanisms. When we planned the experimental design, it was not possible to collect enough individuals in the preclinical phase of T1D. We are planning a follow-up study which will include children in the preclinical phase with detectable autoantibodies to islet cells, identified in a nation-wide screen, which was recently established.

5. The difference between experiment 1 and experiment 2 is unclear.

We thank the reviewer for highlighting this important point. To address this and related comments, we have substantially revised the manuscript to clarify the differences between the initial and final experiments. The initial experiment (batches 1–2) included all CD4⁺ or CD8⁺ T cells, while the final experiment (batches 3–6) was enriched for non-naïve cells (1:5 naïve:non-naïve ratio). We now explain this clearly in the Results and Methods, and added Fig. S3 and Table S3 to illustrate the sorting strategies and batch composition.

6. Genes present in less than 3 cells were removed. It is more common to see this threshold set to 5-10% of cells, in order to remove effects of rare genes.

We thank the reviewer for raising this point. Our decision to retain genes expressed in at least 3 cells aligns with established best practices in single-cell RNA-seq analysis. For instance, Cheng et al. (STAR Protocols, 2023) recommend the filtering threshold 3–5 cells as a strategy to balance noise reduction

with retention of rare-subset markers. The official guides of Seurat [link] likewise suggest filtering genes expressed in at least 3 cells. Setting a threshold based on percentage (e.g., 5–10%) would be too stringent for our heterogeneous T-cell dataset, leading to the loss of biologically significant genes.

7. In the methods, it is stated that cells with <200 genes are removed, and in another place it is stated that cells with <500 cells are removed.

We thank the reviewer for noticing this. First, we removed cells with <200 genes when creating the initial Seurat objects from each batch (as per standard Seurat guidelines; code available in GitHub folder code01_raw_to_init). In a subsequent QC step of the merged dataset, after inspecting the actual violin plots with quality metrics (code in GitHub folder code02_analysis_of_init, parts 02 and 03), we applied a more stringent filter and removed cells with <500 genes. We have clarified this in the Methods section to avoid confusion.

8. What criteria were used to select 12 PCs?

We selected 12 PCs based on the scree plot generated using the Seurat function ElbowPlot.

9. What is the justification for using the GSE11057 and GSE9650 databases for GSEA rather than a more standard database?

We thank the reviewer for this question. We chose GSE11057 and GSE9650 because they are high-quality, well-established datasets with clear experimental designs, and their gene signatures are included in the Molecular Signatures Database (MSigDB) curated collection (C7 category). We are not certain which “standard database” the reviewer refers to; however, many pathway databases (e.g., GO, Biocarta, Reactome) often include relatively small gene sets, which can reduce statistical power in GSEA analyses.

10. Why were fastq files merged prior to HLA typing?

The fastq files were merged to generate a single file per patient for practical and analytical reasons. The HLA typing algorithm aims to determine the individual’s germline HLA genotype, which is identical across all cells. Therefore, maintaining single-cell resolution is unnecessary for this step. Moreover, as the HLA typing software estimates allele probabilities based on read counts mapping to different HLA alleles, merging data from all cells of a given donor increases the overall coverage and maximizes the output accuracy.

11. No justification is provided for the focus on BTN3A2. Why was this gene scaled differently than the others? The use of a range of -1 to 1 for scaling BTN3A2 seems arbitrary.

We thank the reviewer for this important comment and apologize for not making our rationale clear enough in the original manuscript. Our focus on BTN3A2 was motivated by two main reasons:

- 1. Genetic context: BTN3A2 is located in the extended MHC region. Thus, SNPs regulating its expression can be linked to specific HLA haplotypes, which are highly relevant in T1D pathogenesis.*
- 2. Unexpected expression pattern: A previous study (PMID: 31311800) reported upregulation of BTN3A2 (and to a lesser extent BTN3A3) in T cells from children with T1D compared to partially*

HLA-matched controls. In contrast, we observed higher expression of BTN3A2 in the children without T1D in our dataset (Fig. 2A-B), prompting us to explore this discrepancy further.

Regarding the scaling, we compared BTN3A2 expression across multiple studies that used different methods (scRNA-seq of sorted T cells, scRNA-seq of PBMC, bulk RNA-seq of sorted T cells, and whole blood RNA-seq). To allow for consistent cross-study comparisons, we standardized expression values to a range of -1 to 1 within each dataset. This approach avoids biases introduced by differences in data types, sequencing depth, and normalization methods, and was applied uniformly across all genes when comparing datasets (including BTN3A2). We have clarified these points in the revised Results and Methods.

12. In the methods for flow cytometry, the authors state that the same number of cells was used in each batch, but they also state that the same number of cells were used and these numbers look quite different.

We thank the reviewer for this comment. We apologize for the confusion. While our goal was to use a comparable cell number across batches, the difference occurred due to sample availability (1.2 million in batch one, 2 million in batch two). Additionally, final percentages of cell subsets can vary due to differences in cell viability and frequencies during gating. We have clarified this in the Methods section of the revised manuscript.

13. The paper is written in a somewhat non-standard way, with text relevant to the methods included in the results, and very few summary statistics in the text of the results which makes it difficult to evaluate the statements made.

We thank the reviewer for this comment. We acknowledge that preferences for including methodological detail and summary statistics in the Results section can vary. Some other reviewers requested even more methodological information be included directly in the Results, and our experience with immunology journals suggests that detailed summary statistics are often primarily presented in figures to maintain readability. However, we defer to the editor's guidance on this point and are happy to adjust the manuscript format as recommended.

14. Which prior distributions were used in the Bayesian modeling?

We used the default priors in the brms package, specifically: i) The intercept for the mean has student-t prior with 3 degrees of freedom and scale 2.5, with location guessed by brms from the overall range of the data (1.81 for CD4 data and 1.97 for CD8 data), ii) Student-t prior with 3 degrees of freedom and scale 2.5 centered at 0 for the intercept for overdispersion, iii) Improper flat prior for all fixed effects, iv) Half Student-t prior with 3 degrees of freedom, scale 2.5 and location 0 for standard deviation of random effects, v) LKJ(1) prior for correlations between random effects. We included this information in the Methods section of the revised manuscript.

15. In the Figure 1D and 1G heatmaps, it could be useful to present the genes in the same order. Dot plots that show both the average level of expression and the percent of cells in which the genes were expressed could be informative. How were these subsets of genes selected?

We thank the reviewer for this suggestion. The genes shown in Fig. 1D and 1G represent different marker sets specific to each cell type, and they are ordered by expression pattern to illustrate the rationale behind cluster annotation. These genes were selected as established markers of the respective subsets, and the heatmaps are intended primarily for annotation and validation purposes. We believe this

presentation is sufficient and clear; additionally, our heatmap style is inspired by figures from this and related journals, such as PMID: 31624246 (Nature Communications) or PMID: 35978192 (Nature).

16. Figure 2a and 2b appear to be a volcano plot that incorporates both the T1D T0 vs Healthy comparison and the T1D T1 vs Healthy comparison, but this figure is not explained very well. Figure 2c could also use additional explanation.

We thank the reviewer for this helpful comment. We would like to clarify that the plots in Fig. 2A are not classical volcano plots, but rather 2D scatter plots where each gene is plotted by its log-fold change in T1D T0 vs. healthy donors (x-axis) and T1D T1 vs. healthy donors (y-axis). We agree that additional explanation was needed and have revised both the figure legend and the manuscript text to clarify this.

17. The authors claim that this study is the first analysis of repeat measures of single cell transcriptomics in T cells in newly diagnosed patients. At least one other study (PMID: 38967999) seems to meet these criteria?

We thank the reviewer for pointing out this article. We have now cited it in the revised manuscript. However, this study does not fully meet the criteria we described: it did not include longitudinal single-cell transcriptomic or other omics analyses (only flow cytometry), and their BD Rhapsody analysis was limited to a small cohort (5 T1D donors and 5 controls) focusing solely on CD8⁺ T cells, and thus, excluding CD4⁺ T cells.

REVIEWERS' COMMENTS

Reviewer #1 (Remarks to the Author):

We are generally satisfied with the authors' responses, except for their response to minor concern #2. The authors identified a contaminating NK cluster within the Temra cluster (Fig. 5F) in the original manuscript. We suggest the authors remove the NK cluster (Fig. 5F) during quality control rather than in the final analyses because the NK contamination may bias all the results in fig 5. Surprisingly, the authors only removed the NK cluster in Fig 5F while Fig5A-5G remained unchanged. We still believe that all panels in fig 5 should be remade after removing the NK cluster.

We thank the reviewer for their positive feedback and we apologize for the misunderstanding regarding the contaminating NK cluster. We agree with the reviewer that the original Figures 5A-G may have been affected by the presence of the NK-cell contaminants and thank them for highlighting this point.

In the revised manuscript, we have removed the contaminating NK cluster from all analyses of the CD4⁺ Temra subclustering figure (now Fig. 7; corresponding to Fig. 5 in the original submission). The entire figure (panels A–G) was recomputed after excluding these NK cells. We have also updated the Methods to explicitly describe this step. We note that we did not remove NK cells from the global dataset used for the remainder of the manuscript, as doing so would have required reprocessing all analyses and figures without altering any conclusions outside of the Temra subclustering.

Reviewer #1 (Remarks on code availability):

It's OK

Reviewer #2 (Remarks to the Author):

We thank the reviewer for their contribution to the review process.

Reviewer #3 (Remarks to the Author):

The authors have worked hard to improve the quality and clarity of the manuscript. In this process they have addressed most of my concerns.

We thank the reviewer for their positive feedback.

One outlying issue that I feel needs to be mentioned is the HLA-matching. While I agree that 'perfect' HLA matching is not possible, it is possible to match for the HLA alleles of known significance for T1D (HLA-DR4, DQ8, DR3 and DQ2). I also accept that this 'matching' is not perfect it is nonetheless an attempt to avoid being misled by difference attributable to HLA alleles. Not doing this leaves open the

possibility that the differences observed between individuals with and without T1D may simply be attributable to the HLA alleles within each group, not their T1D status.

We thank the reviewer for raising this point. We have now explicitly stated in the Methods that healthy controls were not HLA-matched to participants with T1D (lines 459-460 in the revised manuscript). We also note that we address the potential influence of HLA background on gene expression in our Results and Supplementary Figures.

Reviewer #4 (Remarks to the Author):

Thank you for your revisions which have answered all of my queries except for the question below on raw data availability. Although I would still expect the raw data to be made available I am also happy to defer to the editor on what is the minimum acceptable to the journal.

“Processed data required to generate figures for the manuscript are available on Zenodo: DOI: 10.5281/zenodo.14222418 Raw data were not deposited to protect the identity of study participants.” Under FAIR data principles, I would expect to have access to the raw data upon point of publication e.g. fastq files as much information is lost by running the cellranger pipeline and conversion to a Seurat object. I don’t understand what how this could compromise the identity of study participants? Files could be renamed if they include study ID or participant names. The authors should discuss this with the editor.

We thank the reviewer for this comment. The privacy challenges associated with sharing next-generation sequencing data have been discussed recently in multiple studies and reviews (e.g., PMID: 34759381, PMID: 39362221). Raw sequencing data can be effectively used to identify individual donors based on SNP (PMID: 38622708). Due to European privacy laws (GDPR), individual-level data from living human subjects cannot be made fully public, similar to other studies published in Nature Communications (e.g., PMID: 38714671). Our raw data may be made available to qualified researchers for the sole purpose of replicating procedures and results, provided that data transfer complies with GDPR and is approved by the relevant ethical committees. We defer to the editor’s guidance on data availability requirements.

We thank the reviewer for their positive feedback. In the final version of the manuscript, we have followed the editor’s guidance on data availability requirements.

Reviewer #4 (Remarks on code availability):

As previously, I have checked the code and it looks good but I did not run it myself

Reviewer #5 (Remarks to the Author):

Thank you for the thoughtful responses to my comments. My remaining suggestion would be to include something in the methods about the use of the Bonferroni correction and the fact that study participants were frequency matched rather than 1:1 matched.

We thank the reviewer for their positive feedback. We updated the methods section to mention the frequency matching (lines 458-459 in the revised manuscript) and the use of the Bonferroni correction (lines 543-544 in the revised manuscript).